# On the possibility of a 1000 km long active subglacial water channel under the north Greenland ice sheet

Christopher Chambers[1], Ralf Greve[1,2], Bas Altena[3,4], and Pierre-Marie Lefeuvre[3]

[1]Institute of Low Temperature Science, Hokkaido University, Sapporo, Japan
[2]Arctic Research Center, Hokkaido University, Sapporo, Japan
[3]Department of Geosciences, University of Oslo, Norway
[4]Institute for Marine and Atmospheric research Utrecht, Utrecht University, Utrecht, the Netherlands

**Correspondence:** Christopher Chambers (youstormorg@gmail.com)

**Abstract.** Does an active subglacial water channel, or wet sediment flow, contained within a ~1000 km long valley, with a source at a known area of basal melting deep in the interior of the Greenland ice sheet, reach the sea at the Petermann Glacier grounding line? Basal topographic data shows a segmented valley extending from Petermann Fjord into the centre of Greenland, however the locations of radar scan lines, used to create the bedrock topography data, indicate that valley discontinuity is due to data interpolation. Therefore, as a thought experiment, simulations where the valley is opened are used to investigate its effects on basal water movement and distribution. The simulations indicate that the opening of this valley can result in an uninterrupted water pathway from the interior to Petermann Fjord. Along its length, the path of the valley progresses gradually down an ice surface slope causing a lowering of ice overburden pressure that could enable water and sediment flow along its path. The fact that the valley base appears to be relatively flat and follows a path near the interior ice divide that roughly intersects the east and west basal hydrological basins, is presented as evidence that its present day form may have developed in conjunction with an overlying ice sheet. Experiments where basal melting is increased solely within the deep interior near the known large area of basal melting, result in an increase in the flux of water northwards along the entire valley. The results are consistent with a present day active long subglacial water channel, however considerable uncertainty remains over aspects such as whether adequate water is available at the bed, whether water escapes from the valley or is refrozen, and over what form a hydrological or wet sediment conduit could take along the valley base.

## 1 Introduction

The surface of the Greenland ice sheet holds visual clues to the topography of the bedrock, which in the interior can be below 2 to over 3 km of ice. Ekholm et al. (1998) found two, roughly 75 km long, elongated depressions in the surface of the ice that were connected by a "more than 100 km long, gently curving trench". Ice penetrating radar returns from the depressions were not "mirrorlike", which was considered a possible indication that subglacial water was being transported in the trench northward through a basal hydrological system. With improved topographic data Bamber et al. (2013) identified a "paleofluvial mega-canyon" that extends from central Greenland all the way to Petermann Fjord (Figure 1). The Ekholm et al. features are interior sections of this "canyon". While the feature was referred to as paleofluvial, Bamber et al. (2013) also suggested that the

valley could have water flowing through sections of it today. Specifically they demonstrated water was routed along independent sections of the valley but not along the whole length. In situ observations of water in the valley have not been obtained to date, nor are there current plans to acquire them. Since this "trench" (Ekholm et al., 1998), "subglacial valley" (van der Veen et al., 2007) or "canyon" (Bamber et al., 2013) takes a variety of cross-sectional forms along its length, in this article we will simply refer to it in the broadest term as a subglacial "valley".

The BedMachine v3 basal topographic dataset (Morlighem et al., 2017) shows that the valley appears to be blocked by topographic rises at many points along its route (Figure 1b,c). However, based on the locations of the radar data lines that were used to generate this dataset and the limited extent of the valley bed elevation derived by mass conservation (two example regions are shown in Figure 1c,d), it is clear that these rises occur only in regions where data was not obtained. The results of Bamber et al. (2013) showed water routing along numerous independent sections of the valley, however they inferred that water was being routed away from the valley in these data sparse regions and so the valley was "likely to have influenced basal water flow from the ice sheet interior to the margin". Since these rises are due to kriging interpolation, there is currently no evidence to suggest that this valley is filled (see Appendix A for further detail on BedMachine error estimates). This poses the question; are these rises damming subglacial water flow along this conduit and negatively impacting ice sheet model simulations?

If it is assumed that the valley is open, then the elevation of the bottom of the valley can be roughly determined by the points along its route where data was obtained. In (Figure 1b) The gaps in the valley where the valley base elevation rises above $-100\,\mathrm{m}$ occur where no data has been obtained and interpolation has smoothed out the valley. In fact, after accounting for the smoothing effects of interpolation, a roughly level incised valley will only be resolved correctly exactly at the points where the data was obtained and everywhere else it will be shallower than it should be. Taking this into account a rough assessment is that the valley has a base that varies between $-250\,\mathrm{m}$ and $-500\,\mathrm{m}$ along its length from Interior to Petermann in Figure 1b.

The term "subglacial river" has been previously used (Mooers, 1989; Clayton et al., 1999; Remy and Legresy, 2004; Popov and Masolov, 2007; Klokočník et al., 2018), to describe under ice sheet subglacial hydrological drainage within a valley. Despite this, there appears to be some resistance to the use of the term "subglacial river" in the field of ice sheet subglacial hydrology. The term "active subglacial water channel" could perhaps be interchanged with "subglacial river" if this term is accepted fully by the field in the future, if it is not already today. These terms are used to cover a variety of possible non-film subglacial hydrological conduit forms such as within R-channels, canals, or Nye channels. These different forms are explained further in the discussion. In addition, an "active subglacial water channel" may incorporate storage within reservoirs along route that release water only over certain periods, and can flow uphill in certain situations. An "active subglacial water channel" beneath an ice sheet is therefore considered to have quite different properties from those of a terrestrial river.

The goal of our research is to investigate the impact of a continuous subglacial valley on the flow of basal water as a thought experiment, using a state-of-the-art ice sheet model. In addition, the effects on ice sheet sliding are explored as well as the impact of focussed interior basal melt on water flow along the valley. This is done in the framework of investigating whether a present-day active subglacial water channel along the valley base is possible.

## 2 Model and methods

### 2.1 Spin-up until 1990

We use the SImulation COde for POLythermal Ice Sheets (SICOPOLIS, www.sicopolis.net) version 5.1 (Greve, 2019b), a polythermal ice sheet model originally created by Greve (1995, 1997). To simulate the state of the Greenland ice sheet for our reference year 1990, we carry out a spin-up over the last glacial/interglacial cycle (134 ka). The main forcing is the surface temperature anomaly derived from the $\delta^{18}O$ record of the NGRIP ice core (Nielsen et al., 2018), modified by a surface temperature anomaly derived for the GISP2 site for the final 4 ka (Kobashi et al., 2011). Except for the topographic and geothermal heat flux sensitivity tests described below, the set-up is identical to the one employed for the Ice Sheet Model Intercomparison Project for CMIP6 (ISMIP6), which, in turn, is based on the one used by Greve (2019a). It will be described in detail elsewhere (Goelzer et al., 2020; Greve et al., 2020) and shall only be summarized here.

During the last 9 ka, the horizontal resolution is 5 km, and the computed topography is continuously nudged towards the (slightly smoothed) observed present-day topography. Prior to 1 ka ago, this is done by the method described by Rückamp et al. (2019), and shallow-ice dynamics is employed. For the last 1 ka, nudging is achieved via the 'implied SMB' by Calov et al. (2018) with a relaxation time of 100 a, and hybrid shallow-ice–shelfy-stream dynamics is used (Bernales et al., 2017). Ice thermodynamics is treated by the one-layer melting-CTS enthalpy method (Greve and Blatter, 2016). The bed topography is BedMachine v3 (Morlighem et al., 2017), the geothermal heat flux is by Greve (2019a), and glacial isostatic adjustment (GIA) is modelled by the local-lithosphere–relaxing-asthenosphere (LLRA) approach with a time lag of 3 ka (Le Meur and Huybrechts, 1996). For the topographic and geothermal heat flux sensitivity tests (Sects. 2.3, 2.4), only the last 9 ka of the spin-up are re-computed.

### 2.2 Basal sliding and hydrology

Following the approach by Goelzer et al. (2020) and Greve et al. (2020), we use a basal sliding law that incorporates basal hydrology. The hydrology model is coupled to the ice dynamics using a modified Weertman-Budd-type sliding law proposed by Kleiner and Humbert (2014) with the parameters determined by Calov et al. (2018). The flux and storage of water in the subglacial hydrology model is governed by both the water pressure and the "elevation potential" which when considered together is known as the hydraulic potential (Shreve, 1972; Le Brocq et al., 2006). The basal melt rates from SICOPOLIS are used as the water input for the routing scheme and there is no basal water source from ice sheet surface melting. Water moves in a layer only a few mm thick as a distributed water film where the water pressure and ice overburden pressure are in equilibrium. As such there is no explicit subglacial channelized water flow in the current formulation and so the thin-film model is used as a guide for where subglacial water is moving and collecting under the ice sheet. Where the film is thickest along an uninterrupted path to an ocean entry is considered to be a sign of a basal environment with an increased likelihood of consisting of some form of active subglacial water channel. The flux-routing method requires that local sinks and flat areas are removed, and this is done using a Priority-Flood algorithm which fills depressions and adds a small gradient, using the method of Barnes et al. (2014) to a depth of 10 m to account for subglacial lakes.

The basal sliding coefficient is determined individually for 20 different regions, the 19 basins by Zwally et al. (2012) plus a separate region for the Northeast Greenland Ice Stream (NEGIS, defined by $\geq 50\,\mathrm{m\,a^{-1}}$ surface velocity). This is done iteratively for the last $1\,\mathrm{ka}$ of the spin-up sequence by minimizing the RMSD between simulated and observed logarithmic surface velocities. A detailed description of this procedure will be given elsewhere (Greve et al., 2020). Note that we do not re-

compute the optimization for the topographic and geothermal heat flux sensitivity tests (Sects. 2.3, 2.4), rather, the coefficients determined by the standard bed topography (Morlighem et al., 2017) and geothermal heat flux (Greve, 2019a) are used for all tests unchanged.

## 2.3    Bedrock modifications

The bedrock topographic data is altered in a way to ensure that the valley is open from Interior to Petermann in at an average

depth of around $-400\,\mathrm{m}$ (Figure 2b). The case with the standard topography is referred to as "Control" and the case with the open valley as "Valley". The simulation initialisations are otherwise identical so that the results only show the consequences of the topographic change.

The topographic modification is done using a flow-oriented interpolation scheme. Given the BedMachine topography and flight lines (Figure 1), a polygon of the rough location of the valley is drawn using a geographic information system (GIS).

A buffer of 200 kilometres around this polygon is created and measurements falling into this buffer, but not into the valley polygon, are used to interpolate a new BedMachine, with the same procedure and parameters as Morlighem et al. (2017). The polygon outlining the valley is converted to a centerline, and then described through a spline. Data points situated within the valley polygon are converted from their Cartesian coordinates $(x, y)$ towards this flow-oriented coordinate $(s, n)$ system. This is a common reference frame, where $s$ describes the distance of the thalweg and $n$ is the normal in right-hand direction,

and it outperforms ordinary kriging and others when used in with an anistropic adjustment (Merwade et al., 2006). Here the anisotropic factor was 10, with a nugget of 20 metres, a sill of 30 metres and a range of 300 metres. Further details on the procedure are detailed in Legleiter and Kyriakidis (2008).

For three regions, saddles are present along the valley. Bounding boxes over these saddlepoints are drawn that cover both the saddle and trenches within. Over these subsets a watershed algorithm called Maximally Stable Extremal Region (MSER;

Donoser and Bischof, 2006) tracking is run to detect the trenches to be connected. Pixels between both trenches of the saddle are found through connecting mathematical morphologic operations. These selected pixels are then adjusted by linear interpolation of the elevation information in the trenches to make a seamless passage.

## 2.4    Idealized interior basal melt sensitivity experiments

Two additional sensitivity tests are designed to assess where basal water melted near the source of NEGIS is routed. The tests

are referred to as ControlS that uses the standard topography as before, and ValleyS that uses the Valley topography. Whereas the geothermal heat flux distribution of Greve (2019a) is used in Control and Valley, for ControlS and ValleyS the geothermal heat flux is increased locally to generate a basal melt rate of between 0.13 to $0.14\,\mathrm{m\,a^{-1}}$ near the source of NEGIS as shown in Figure 7b. To do this a bell-shaped geothermal heat flux anomaly centred at 74°N, 40°W is introduced with a 1-sigma radius

of 50 km and peak heat flux of $1.5\,\mathrm{W\,m^{-2}}$. The anomaly is located over the region of enhanced basal melting at the source of NEGIS shown in Fahnestock et al. (2001, Figure 2). The intention here is not to produce a realistic melt rate distribution but rather to test the effect of increasing melting in the interior near the source of NEGIS. However, both the maximum melt rates, and the cross-sectional distribution are comparable to the cross section of derived basal melt rates of Fahnestock et al. (2001, Figure 3).

## 3 Results

All results presented here are from the SICOPOLIS simulation output for the year 1990 at 5 km horizontal resolution as detailed above. To examine the effect that the introduction of an uninterrupted valley has on the simulated ice sheet, an analysis of the basal water depth, basal water flux, and ice sheet velocity are presented.

For north Greenland the simulated basal water depth is affected by the introduction of the valley in several ways. In Figure 3a it can be seen that the standard topography produces independent areas of deeper basal water along the valley but there are clear gaps between these areas. In contrast, when a continuous valley is introduced (Figure 3b) the basal water depth is both deeper and uninterrupted along the length of the valley all the way to Petermann Fjord. The thickest water depth ($> 0.01\,\mathrm{m}$) along the valley route occurs where the Priority Flood algorithm has been activated to represent subglacial lakes. In most interior areas away from the valley there is little or no change in the basal water depth. In particular, there is little to no effect on the basal water pathways associated with NEGIS. The interior basal water changes are relatively small because the valley follows a path close to the boundary between the east and west basal water catchments and thus has less influence on them.

To obtain a clearer picture of the changes to the basal water due to the introduction of the valley, the difference in basal water between the cases (Valley – Control) is shown in Figure 3c. Doing this reveals that the increased water within the valley is surrounded predominantly by a reduction in basal water adjacent to the valley. Basal water reduction extends to some regions away from the valley, particularly to the west of the interior section. The one region of increased basal water outside of the valley is a region that extends towards the Petermann Ice Stream at NEEM zone in Figure 3d. The effect of these changes in the Petermann catchment is to redistribute the basal water into a narrower and deeper plume that can also be seen in Figure 3b.

To examine how the movement of basal water is altered by the introduction of the valley, the basal water flux is presented in Figure 4. The valley causes a shift in basal water flux in its near vicinity, with increased flux within the valley base. Water flux streamlines give an indication that water flux is generally down-valley with streamlines getting "stuck" in the valley along certain sections. In the Petermann catchment region, the increase in water flux in the valley causes a shift downstream in the subglacial water distribution where the valley crosses a region of increased flux out of the interior (NEEM zone in Figure 4). Along this section where the valley is oriented SSW to ENE the flux just NNW of the valley is reduced in the south and then increased to the north in the region upstream of the interior part of the Petermann Ice Stream. This is consistent with the increase in basal water mentioned above. The simulated effect of the valley is therefore to focus maximum water flux into a narrower but more elongated region that is also shifted eastward. Sensitivity tests (Appendix B) indicate that the location and magnitude of this water flux out of the valley is sensitive to the valley depth due to its consequent effect on the steepness of

the valley sides. Figure 4b (lower panel) indicates a basal water flux of $10,000\,\mathrm{m^2\,a^{-1}}$ in the valley across 5km grid boxes upstream of NEEM zone. This corresponds to a discharge of just $1.6\,\mathrm{m^3\,s^{-1}}$.

The rule of thumb helpful for understanding the relative roles on subglacial water flow of ice overburden pressure and basal topography, is that the topographic gradient needs to be 11 times greater than, and opposing, the ice surface slope for water flowing along the bed to start accumulating (e.g., Cuffey and Paterson, 2010). Figure 5 indicates that the along-valley component of the ice surface gently slopes down-valley all the way to Petermann Fjord. This is an indication that the ice overburden pressure distribution does not oppose the flow of water towards the north and should generally reinforce it for the northern half of the introduced valley. As the ice sheet surface is very gently sloping in the interior, the basal topography and water fluxes will have a greater influence on subglacial water routing than around the edges of the ice sheet. In this situation the valley base topography could be either sloping downward towards the north, or flat, for water to flow northward. It appears it could be closer to the latter, which is consistent with subglacial erosive and depositional water activity but does not prove it. The isostatic correction in Bamber et al. (2013, Figure S5) implies that for an ice-free Greenland the valley would be $600\,\mathrm{m}$ higher than present in the interior progressing to 200 metres higher near the coast. Today the valley base appears to be consistently between $-250$ and $-500$ metres with no clear trend along-valley.

To demonstrate the influence of the valley on simulated ice sheet sliding, Figure 6 shows the ice surface velocity difference between Valley and Control to highlight the locations where Valley increases or decreases sliding. The sliding changes are relatively modest and localized, with only small regions at certain outlet glaciers having a greater than $10\,\mathrm{m\,a^{-1}}$ change. Some sliding changes occur in the Petermann catchment (Figure 6b) where sliding increases over the valley and very weakly ($\sim 0.5\,\mathrm{m\,a^{-1}}$) in the Petermann Ice Stream. These increases are consistent with the redistribution of basal water seen in Figure 3. The sliding changes should be viewed as demonstrations of the potential for change due to the introduction of an open valley while considering that the simulated basal hydrology is limited by its reliance on a thin-film model. Small sliding changes occur at certain outlet glaciers such as Ryder Glacier and several along the west coast. These changes, away from the valley and Petermann, are inconsistent across different model setups. Considering other uncertainties and inaccuracies, the differences are too small to allow assessing which case is better compared to observed surface velocities.

To investigate whether water is transported down the length of the valley, two additional sensitivity simulations have been completed as described in section 2.4. The simulations compare scenarios with and without the open valley that also include an area of enhanced interior basal melting as shown in Figure 7a,b. Figure 7c shows the change in basal water depth that occurs when you introduce the enhanced basal melting region into simulations with the standard topography. The greater basal melting generates larger amounts of basal water that is mostly transported down under NEGIS and adjacent regions. A lesser amount of the extra basal meltwater is transported towards the west coast. The same comparison is made in Figure 7d, however this time the two simulations compared both have an open valley. Comparing Figure 7c with d, the basal water distribution is similar down NEGIS, is reduced towards the west coast, and is increased along the valley down to Petermann, with the two paths down the Petermann Ice Stream and down the valley, evident in the lower reaches. This result demonstrates that simulated meltwater generated solely in the deep interior can be transported down the entire length of the valley and it is notable that it is only at Petermann where there is any significant change to the basal water depth across northern Greenland north of $80°\mathrm{N}$. The other

consequence of this is that the down-valley basal water flux upstream of the NEEM zone increases from $\sim 10{,}000\,\mathrm{m^2\,a^{-1}}$ in Valley to $\sim 50{,}000\,\mathrm{m^2\,a^{-1}}$ in ValleyS which corresponds to an increase in thin film discharge across $5\,\mathrm{km}$ grid points from $\sim 1.6\,\mathrm{m^3\,s^{-1}}$ to $\sim 7.9\,\mathrm{m^3\,s^{-1}}$.

## 4 Discussion

The formation of subglacial water channels has long been known to be a fundamental evolutionary property of subglacial water flow. Röthlisberger (1972) and Shreve (1972) proposed that subglacial water can form channels that cut upwards into the ice. These have come to be referred to as "R-channels". Channeling of subglacial water occurs because the initial film of water at the ice base can become unstable due to viscous dissipation which initiates the development of R-channels. For this transition to occur the discharge has to increase beyond a threshold (Schoof, 2010).

The R-channel theory requires a hard bed and therefore ignores potential bed erosion from such a channel. If the channels are over a sufficiently hard bedrock and move position then this assumption should hold, however if they remain quasi-stationary, due to basal topography or persistent ice overburden pressure distribution influences, then the effects of bedrock erosion or sediment deposition should manifest. In the case where there is sufficient sediment deposition in a stationary channel an esker could develop lifting the water channel above the bedrock. In the case where there is not sufficient sediment deposition,

erosion downwards into the bed will inevitably occur if the channel remains stationary. Nye (1973) suggested that channels incised upwards into the ice are more vulnerable to closure due to ice overburden pressure and ice movement and concluded that channels incised into the rock were expected to be much longer-lived than channels incised upwards into the ice. Nye concludes that "while there may be temporary channels incised upwards into the ice, there will be comparatively permanent channels cut downwards into the rock bed."

There are other reasons to suppose that flowing water in a subglacial valley could be a favourable mode of water transport under an ice sheet. These are associated with the resistance to freezing of water in such a channel. Firstly, because the ice sheet surface will likely not have as pronounced an indentation as an incised valley in the bedrock beneath has, the ice thickness and therefore ice overburden pressure will be higher at the base of the valley than under the ice over the surrounding bedrock. This increases the likelihood of the ice at the base of a valley being at the pressure melting point.

Secondly, an incised valley under an ice sheet will tend to have higher geothermal heat flux at its base and particularly along its sides. This is because of the distortion of isotherms beneath the valley that increases the isotherm gradient and consequently also the heat flux (e.g., van der Veen et al., 2007). For example, Lees (1910) found that a depth to width ratio of 0.5 increases heat flux by around 50% while van der Veen et al. (2007) found a 100% increase in heat flux in a Jakobshavn-scale idealized simulation. Essentially the deeper and steeper the valley, the greater the heat flux increase will be at the valley bottom. In the

case of a melting ice base, if the valley sides are steep enough to overcome any opposing overburden pressure forcing on water flow, then meltwater will collect at the base of the valley which could also further enhance melting there.

  Thirdly, flowing water generates heat through frictional heating, increasing the temperature of the water. Sediment movement will also generate frictional heat. Water and sediment flow can also transfer heat downstream so factors such as the upstream

water heat capacity and the duration of cooling will determine whether the water freezes or not. Fourthly, applicable to all basal water, is one of the odd properties of water. As water cools below $4°C$ it starts to become less dense causing the coldest water to rise up. Thus, freezing occurs at the top, which in the scenario of an active subglacial water channel would be at the base of the ice sheet. This new ice acts to insulate the liquid water below as is observed in frozen rivers and lakes. This allows liquid

water to persist beneath the ice in situations where it would not if water did not have this property.

These are some factors that may enable flowing water and sediment to continue in a valley under an ice sheet even in some situations where the ice sheet is frozen to the bed outside of the valley. These factors are presented here as indicators that positive feedback processes may exist that favour the development of subglacial water channels incised into the bed. The possibility remains that the present day valley form developed as a consequence of erosion under some or all of the following

1) under current conditions, 2) under last glacial maximum conditions, 3) during ice sheet retreat, 4) under reduced ice sheet cover, and 5) under ice free conditions. We simply do not have enough information. Tunnel valleys provide one demonstration of how subglacial water erosion can erode into hundreds of metres of bedrock. Given that the source is close to a proposed geothermal warm spot, past episodic subglacial down-valley discharges of water and sediment are a possibility. Much smaller amounts of erosion and deposition would be needed to maintain a base slope favourable for water routing, as is typical of rivers

in general.

The model results indicate that the valley follows a path down a gentle ice surface slope (Figure 5) which would imply that the ice overburden pressure lowers as the valley progresses towards Petermann Fjord. In this scenario, if a water channel were to be maintained along a relatively flat uninterrupted valley base, the overburden pressure should propel water towards the ocean outlet. This is providing that water does not escape out the sides of the valley as appears to happen to some of the water

as the valley crosses NEEM zone. If down-valley water propulsion occurs, a possibility is that it does so sporadically through the build-up, and release, of water in reservoirs along the valley route.

The results also indicate that the course of the valley in the interior runs in the vicinity of the boundary of the east and west subglacial hydrological catchments (Figure 3a,b and shown schematically on Figure 8). This catchment boundary occurs in this region because it is below the gentle northward ridge of highest surface ice (Figure 5) which broadly forces the division

between these hydrological basins. A consequence of this positioning is that the valley enters the Petermann surface catchment at its southernmost location (Figure 1a). Because the gentle ice ridge is the region where water flux directed towards the east or west is at a minimum, it represents the most favourable location in the interior of northern Greenland for the development of a hydrological pathway directed towards the north. This possible relationship to the ice sheet shape could be a further indication that the path of this valley has developed as a consequence of the ice overlaying it. However, while the valley appears to

intersect the east and west hydrological basins in the interior it does not follow the water flux streamlines exactly, particularly as it crosses NEEM zone. South of the Petermann surface catchment the valley tracks roughly parralel, and to the west of, the basal hydrological divide, while Tributary projects towards the east of it (Figure 8). If the bedrock were flat, there would be only one basal water route towards the north and it would be directed exactly along the hydrological divide. Perhaps a subglacial valley perfectly aligned with present day basal water flux streamlines is not to be expected given the long period

required to erode it and the different shape of the ice sheet in the past. As an example Bamber et al. (2013, Figure 3b) indicates

that the interior basal hydrological divide is shifted to the west under conditions at the last glacial maximum. Nonetheless the valley still follows a path not far off the basal hydrological divide so it is possible that conditions favourable for subglacial down-valley water routing may have existed for a long time as has already been implied by the results of Bamber et al. (2013).

The current simulations do not include subglacial channelized water flow, such as within R-channels, in the hydrology module. Channelized water flow could funnel different amounts of water away from, and to, particular locations leading to focused areas of suppressed and enhanced sliding. This may effect the sliding model results that should be seen as a view of the potential for change in different regions rather than a prediction for the valley's influence on sliding speed. The valley could extend further southward and there is evidence of tributaries that could increase the main valley's discharge potential. The most prominent possible tributary, shown as "Tributary" in Figure 1 projects towards a region of higher basal melt associated with NEGIS (Fahnestock et al., 2001) and could therefore be an additional source of basal meltwater. Also the main valley appears to continue past Interior by taking a sharp turn towards the east. This directs it to begin at an area of enhanced basal melt associated with the very source of NEGIS shown in Fahnestock et al. (2001, Figures 2 and 3). If this is an active subglacial water channel, then this location seems the most probable source of the majority of the discharge down the valley given the frozen or more slowly melting ice base elsewhere in the valley catchment. A distinct source at a region of high basal melt is also more consistent with subglacial water erosion rather than erosion prior to ice sheet inception when ice would not be available to melt and water sourced from precipitation would presumably be spread across the entirety of Greenland.

The valley originates from under some of the thickest and highest ice in Greenland (Figure 5). The valley we have inserted in the simulations has its upper end at "Interior" but given the basal topographic basin at "Basin" (Figure 1), could it be possible that basal water and sediment is transported from Basin to Interior? Between these two regions the ice surface slope is relatively flat so water flow should be more heavily influenced by the basal topography. In this inter-basin region the basal topography is poorly resolved and it is unknown whether the ice sheet is frozen to the base. It is therefore unclear whether a basal water connection could exist between Basin and Interior. A smaller channel is evident on most, but not all, flight lines that passed over this region (Figures 1b, 5 and A1). In the simulations presented here, Basin is frozen at the ice base and so no basal water is produced there (Figure 3a,b). This is due to the geothermal heating distribution used by SICOPOLIS which is, as with all Greenland geothermal distribution estimates to date, highly uncertain due to severely limited observations at the base (e.g., Rezvanbehbahani et al., 2017). If these basins are connected hydrologically, it could significantly extend the catchment of the valley and imply a subglacial water channel over 1400 km long. At present there is not enough data on the bedrock heat flux or topography to know if this is the case and the fact that we are in the dark on such a potentially large feature on the Earth's surface, expresses the importance of observation campaigns that can improve our understanding of the conditions at the bed.

The path of such a long basal valley down an ice surface slope that appears to roughly intersect the east and west basal hydrological basins in the interior could be an indication that this feature has developed over a long period in conjunction with the ice sheet covering it. The alternative is that it eroded due to a paleo-river flowing when the ice sheet was much smaller, or absent. At that time the topography would have been significantly different due to bedrock isostasy. In addition, the water flow would have been governed by gravity when conditions were ice-free. This different water flow environment would mean that it was coincidental that the same valley follows a path that is today favourable for water transport from the deep interior all

the way to the coast under a thick ice sheet. Since the relationships with the ice sheet are not perfect and speculative in nature, the significance, or lack thereof, of this coincidence will need to be investigated further. However, additionally, the apparent flatness of the valley base in the interior where the ice surface is relatively flat, is, just like any other river, the ultimate erosional and depositional form of a long-term active waterway. Due to bedrock isostasy it would, again, seem to be coincidental that a paleo-river system would have a relatively flat base today. One can imagine that a paleo-river valley pushed down by the weight of the ice as the ice thickened would end up today having an uneven base that depended on the evolution of the competing pressures from the ice and crustal rock. As stated earlier, the base today is not perfectly level as it appears to vary between $-250$ to $-500\,\mathrm{m}$ but there is no obvious along-valley trend over its $1000\,\mathrm{km}$ length. Whether a paleo-river valley could end up having a very long fairly level base in this situation is worthy of future investigation. In the absence of adequate direct observations, perhaps the topographic form of the base of this valley could, with further work, help us deduce whether this is an active subglacial water channel (or wet sediment channel) or not.

Estimating the discharge down an active subglacial water channel, that we don't know exists, in an extremely poorly observed environment is a fool's errand. However the following notes on this issue are provided as a thought experiment and to encourage future investigations. The SICOPOLIS simulated thin-film down-valley discharge upstream of NEEM zone is estimated as being $\sim 1.6\,\mathrm{m^3\,s^{-1}}$ with the standard geothermal heat flux (Valley) and $\sim 7.9\,\mathrm{m^3\,s^{-1}}$ with an included hot spot (ValleyS). Only a very rough estimate of the total discharge generated by the hot spot found by Fahnestock et al. (2001) can be made given the limited coverage of their analysis. Based on a rough outline of the detected areas of melt of $0.1\,\mathrm{m\,a^{-1}}$, a value of order $100\,\mathrm{m^3\,s^{-1}}$ can be obtained. The majority of this may flow northeastward under NEGIS, however at least part of the region of consistently highest melt around the interior tip of NEGIS lies beyond the NEGIS basal hydrological catchment. A very rough estimate gives $\sim 30\,\mathrm{m^3\,s^{-1}}$ that could be routed into the source of the valley. If the basal water that lies outside of the NEGIS basal water catchment is not being evacuated from the region, then substantial resevoirs could be continually filling until the time of release. A constant discharge of $30\,\mathrm{m^3\,s^{-1}}$ can build up 2,592,000 $\mathrm{m^3}$ (2,592,000,000 litres) in one day. Alternatively, refreezing to the ice base could reduce or eliminate this small potential discharge. Analyses such as MacGregor et al. (2016, Figure 11) indicate that along most of the valley length, the base of the ice sheet is "likely frozen". However, since water has been inferred to be present from IPR data in a limited number of locations (Ekholm et al., 1998), and unless this water was entirely melted in place, then, based on both Bamber et al. (2013) and the results presented here, some of the detected water likely came from upvalley. As the valley approaches closer to Petermann Fjord, contributions to the discharge from summer surface melting become more likely and have the potential to overwhelm any discharge from interior basal melting. These discharge calculations are limited in many ways beyond those already mentioned, for example there is no accounting for refreezing to the ice base or melting due to channelized water flow.

A potentially important factor when considering the erosional capability and mass volume transported down valley is the role of sediment and ice. Sediment within subglacial water channels could increase erosion rates and frictional heating. Alternatively basal water could become incorporated into eroded sediment enabling the mobilization of sediment flows or the development of porous flow through the sediment. If mobilized, sediment confined in a relatively level valley should also be transported along the valley in the direction of the along-valley component of ice surface slope gradient. Finally there is the role of the ice

that lies within the valley. Basal ice flow could be modified within the valley and move partially, or wholly along-valley. The possible roles of high pressure subglacial wet or liquified sediment transport, as well as of ice, could be considered in future investigations of this valley.

## 5  Conclusions

The Greenland bedrock data indicates that a subglacial valley extends from Petermann Fjord into the center of Greenland. The valley is segmented along its route in the current bed topographic datasets used in ice sheet simulations. The rises occur where data is interpolated to fill in gaps between where radar has obtained reliable data. This suggests that the valley rises are not real. Therefore, as a thought experiment, simulation tests have been completed to investigate the consequences of removing these rises. Opening up the valley in SICOPOLIS simulations causes water to be re-routed leading to localised modest ice sheet

sliding changes. The valley progresses gradually from thicker to thinner ice causing a lowering of ice overburden pressure that could enable water and sediment flow along its path towards the sea. If this is the case, some of the basal water routed to Petermann Fjord may originate from melting of the deepest and oldest part of the ice sheet. When melting is increased only in the deep interior at a known region of basal melting near the source of NEGIS, the simulated discharge is increased down the entire length of the valley. The results show that even small adjustments in the bed topography to include probable

features can have consequences that could affect simulations of the future ice sheet. The possibility is raised of a long active subglacial water channel (or "subglacial river" if the term is accepted by the ice sheet subglacial hydrological community), or wet sediment flow system that is poorly realized in current ice sheet simulations. If this potential hydrological system has formed and/or is maintained due to the presence of the ice sheet, then it is a fundamentally different system that requires a different understanding to that of a paleo-fluvial river valley that eroded prior to ice sheet formation.

## 20  Appendix A: Error estimates

A map of error estimates from BedMachine v3 (Morlighem et al., 2017) shows the variation in error across north Greenland (Figure A1). Errors range from 2 to $\sim 600$ metres with a median of 158 metres along the valley. Bed elevation is improved in the lower part of the Petermann catchment ($< 250\,\text{km}$ in our profile in Figure A2g) as it is derived from mass conservation and from a dense IceBridge campaign (see Figure 1b for an outline of the mass conservation region). The kriging interpolation is

25 applied to the rest of the interior of the ice sheet and thus most of the valley. The kriging algorithm is described in Morlighem et al. (2017) as "The variogram is modeled as a Gaussian function, with a sill of $100\,\text{m}$, a range of $8\,\text{km}$ and a nugget effect of $50\,\text{m}$, to account for uncertainty in ice thickness measurements". The along valley profile (Figure A2g) indicates that our introduced valley is deeper in the interior than in the BedMachine data. The cross-sections along 3 flight lines across the valley (Figure A2a,b,c) indicate the valley sides have similar slope angles on the $5\,\text{km}$ grid to the observed. 3 more example cross-

30 sections (Figure A2d,e,f) in regions of high BedMachine error (away from flight lines) show the consequent failure to resolve the valley.

## Appendix B: Sensitivity tests

The results from four additional simulations are presented here that test the sensitivity of the water routing to the valley base topographic elevation. Three of the tests use 26 linear $10\,\text{km}$ wide idealized valleys to form an uninterrupted valley from Interior to Petermann. The valleys are created using the Matlab function "inpolygon" that sets grid point values within an along-valley rectangle to be a specified value. The lithosphere is then relaxed in a short SICOPOLIS simulation to produce an isostatically relaxed bed topography. Tests are done with inserted idealized valleys at constant maximum depths of $-100\,\text{m}$, $-300\,\text{m}$, and $-500\,\text{m}$ and are compared to a 4th test which uses standard SICOPOLIS topography as a control simulation. The $-100\,\text{m}$ simulation effectively removes most of the segmented valley while the $-500\,\text{m}$ case best represents the slopes of the sides of the valley. The tests use an otherwise identical method to that described in section 2.1.

The subglacial water flux for the four cases is in Figure A3. There are large differences in water routing in and around the valley, between these cases. For a $-100\,\text{m}$ valley (Figure A3f), the northward water flux signature associated with the valley is largely eliminated. If the valley base is lowered to $-300\,\text{m}$ (Figure A3g), increased valley water flux occurs from Interior to NEEM zone where water then appears to be entirely evacuated from the valley into a plume directed towards Petermann. For the $-500\,\text{m}$ case (Figure A3h) the valley water flux is continuous high from Interior to Petermann and the plume out of the valley at NEEM zone is largely eliminated. The results confirm the finding that NEEM zone is the region most prone to water leakage from the valley. The result for a valley base at $-500\,\text{m}$ suggest that the valley side slopes on the $5\,\text{km}$ grid in this case are steep enough to overcome the northwestward directed hydropotential component due to the ice surface slope. From a modelling perspective the results highlight the need to improve the bedrock topography data in the NEEM zone region.

*Author contributions.*  Chris Chambers initiated the study. Chris Chambers and Ralf Greve set up and carried out the numerical experiments with the SICOPOLIS model. Bas Altena performed the primary subglacial topography alteration operation, while the topography for the sensitivity tests was processed by Chris Chambers and Ralf Greve. Pierre-Marie Lefeuvre prepared data, produced the topographic cross-sections, and analyzed the radar flight lines. Chris Chambers interpreted the results and wrote the manuscript with contributions from all co-authors.

*Code and data availability.*  SICOPOLIS is available as free and open-source software at www.sicopolis.net.

*Competing interests.*  The authors declare no competing interests.

*Acknowledgements.*  We thank Kenichi Matsuoka, Andy Aschwanden, Jonathan Bamber and an anonymous reviewer for constructive remarks and suggestions that helped to improve the manuscript. Chris Chambers and Ralf Greve were supported by Japan Society for the Pro-

motion of Science (JSPS) KAKENHI grant number JP16H02224. Ralf Greve was supported by JSPS KAKENHI grant numbers JP17H06104 and JP17H06323, and by the Arctic Challenge for Sustainability (ArCS) project of the Japanese Ministry of Education, Culture, Sports, Science and Technology (MEXT) (Program Grant Number JPMXD1300000000). The research visit of Bas Altena and Pierre-Marie Lefeuvre to Hokkaido University was supported through the CryoJaNo project funded by SIU (Norwegian Centre for International Cooperation in Higher Education) (HNP-2015/10010). The research of Bas Altena was conducted through support from the European Union FP7 ERC project ICEMASS (320816) and the ESA project ICEFLOW (4000125560 18 I-NS). Pierre-Marie Lefeuvre was financed by the European Union FP7 ERC project ICEMASS (320816), the Norwegian Research Council funded CalvingSEIS project (244196/E10) and the Minister of Climate and Environment.

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

**Figures**

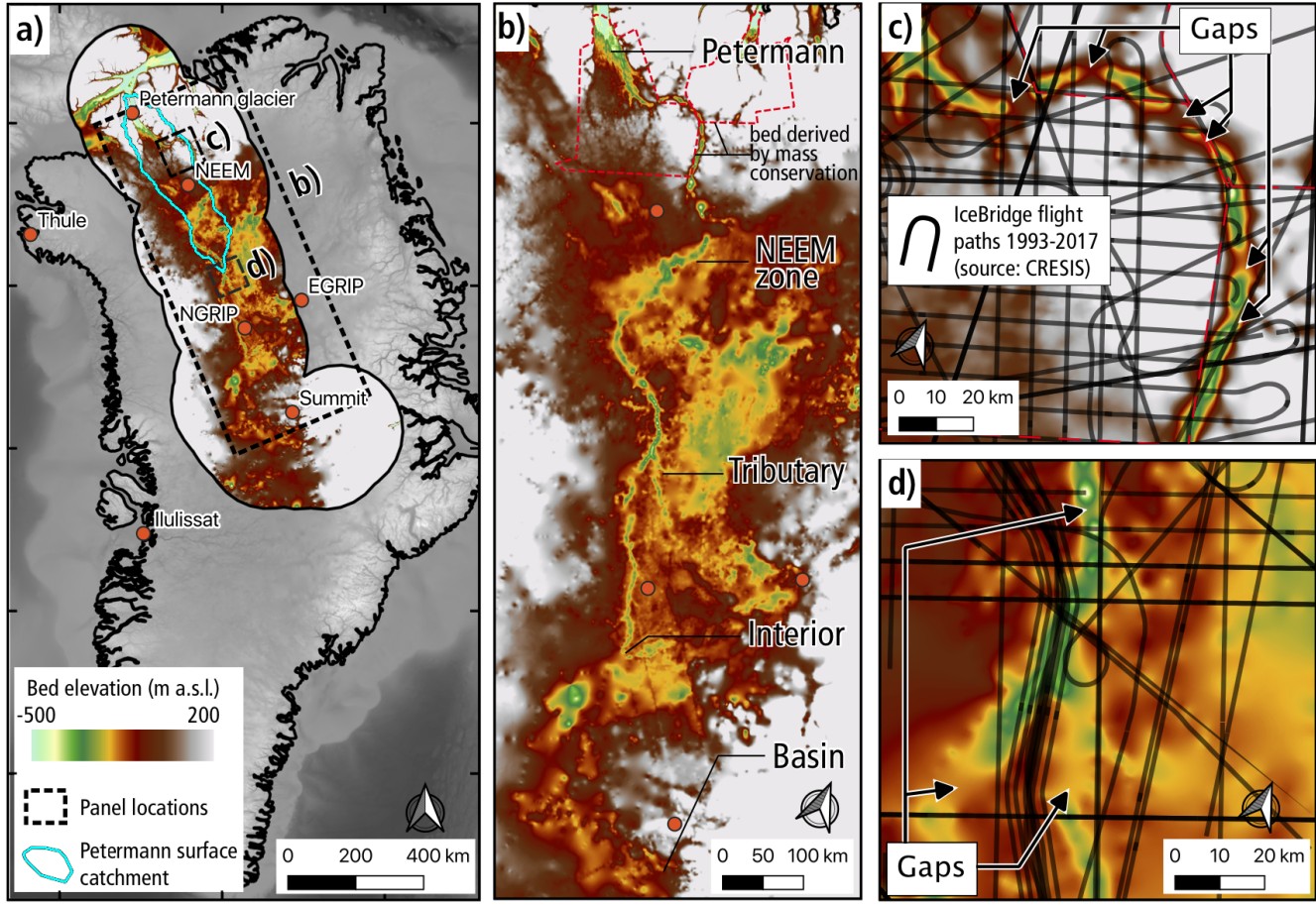

**Figure 1.** BedMachine v3 bed topography (Morlighem et al., 2017) between $-500$ to 200 metres above sea level for a) Greenland overview with boxes for b) the valley region and c) and d), two regions showing the IceBridge flight paths.

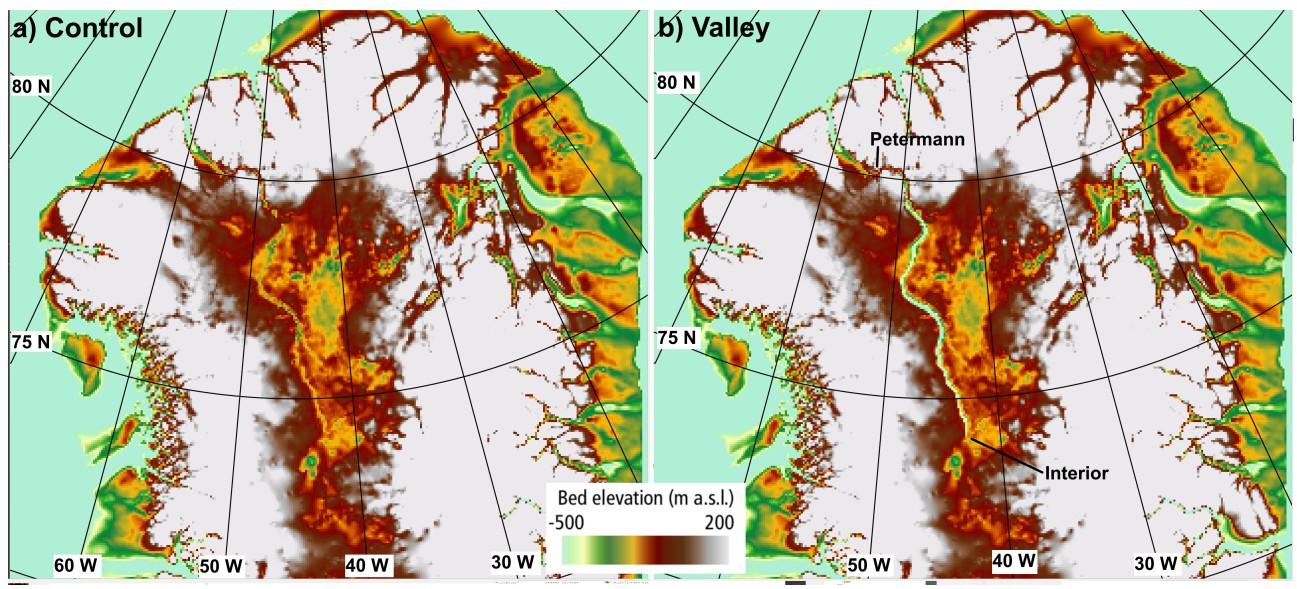

**Figure 2.** Basal topographic height between $-500$ and $200$ metres above sea level for a) Control (standard SICOPOLIS input derived from BedMachine), and b) Valley (manually adjusted from Petermann to Interior).

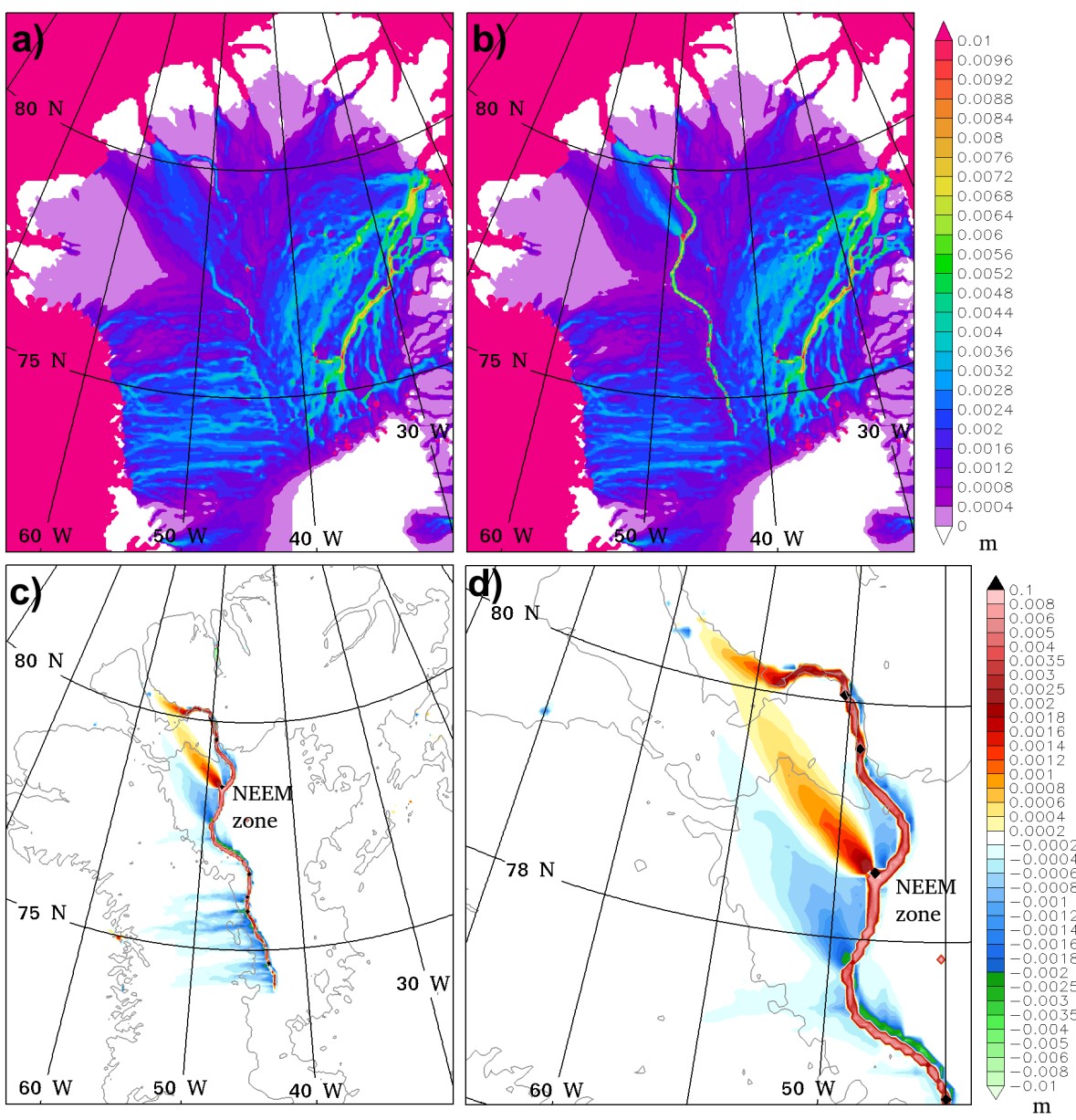

**Figure 3.** Basal water depth (m) for a) Control and b) Valley, from SICOPOLIS simulations for the year 1990. Basal water depth difference (m) (Valley – Control) for c) northwest Greenland and d) Petermann catchment region.

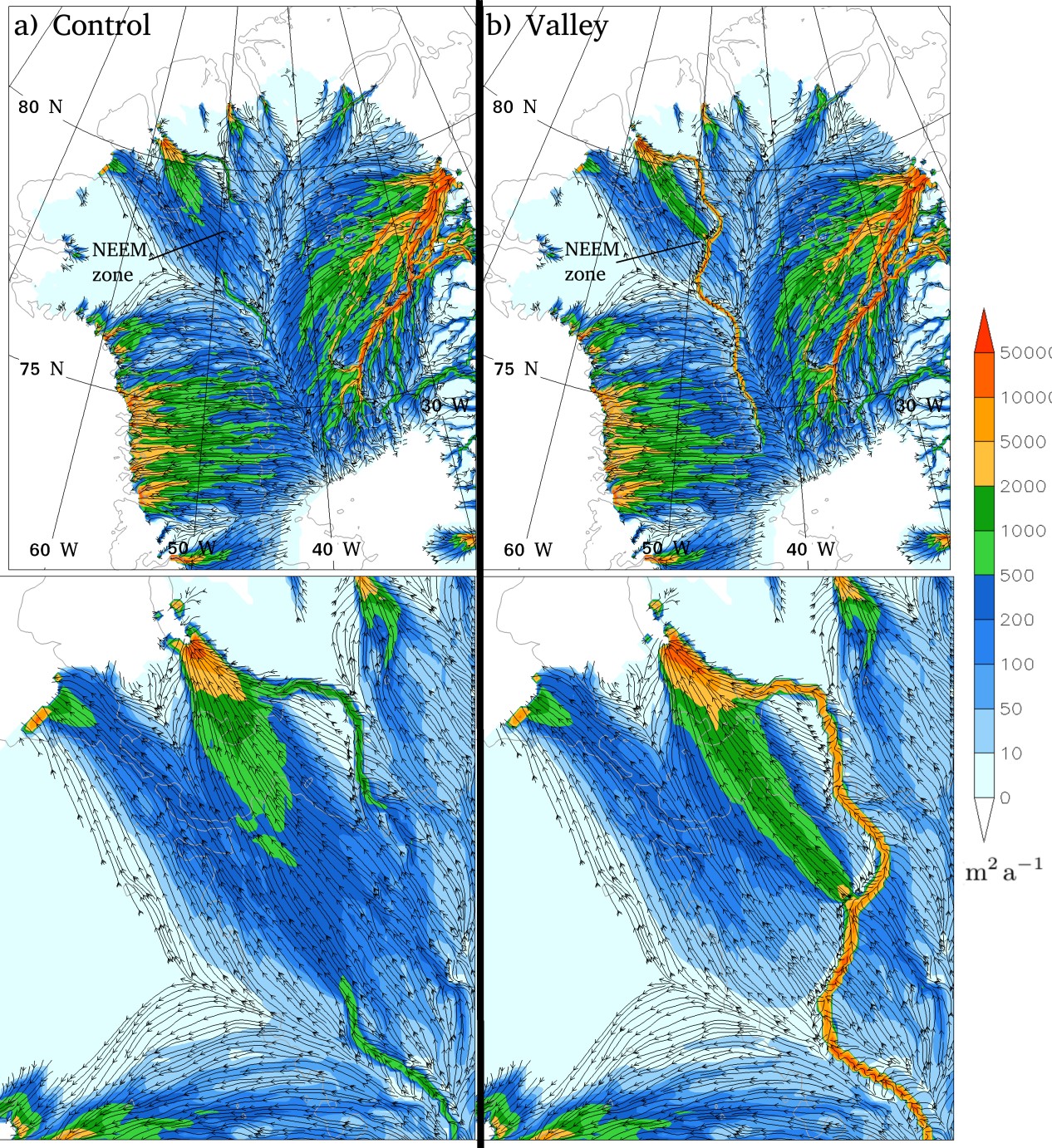

**Figure 4.** Basal water flux magnitude ($\mathrm{m^2\,a^{-1}}$ colours) and streamlines for north Greenland for a) Control and b) Valley. NEEM zone marks where the greatest change occurs out of the valley as discussed in the text and the lower plots zoom into this region for the respective cases above to show detail.

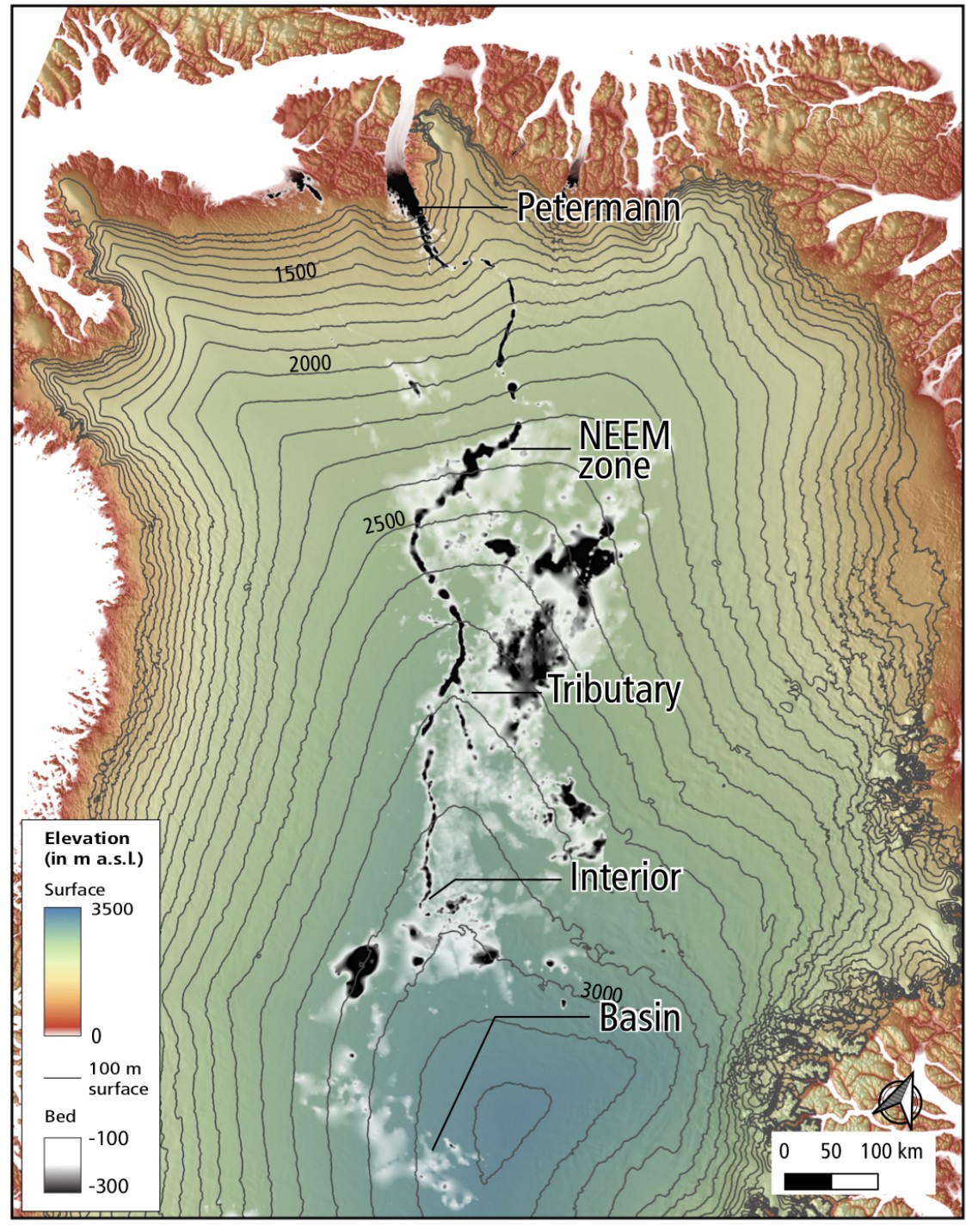

**Figure 5.** Surface elevation (m) with bed elevation for $-100\,$m or lower overlayed in grey to indicate the path of the valley.

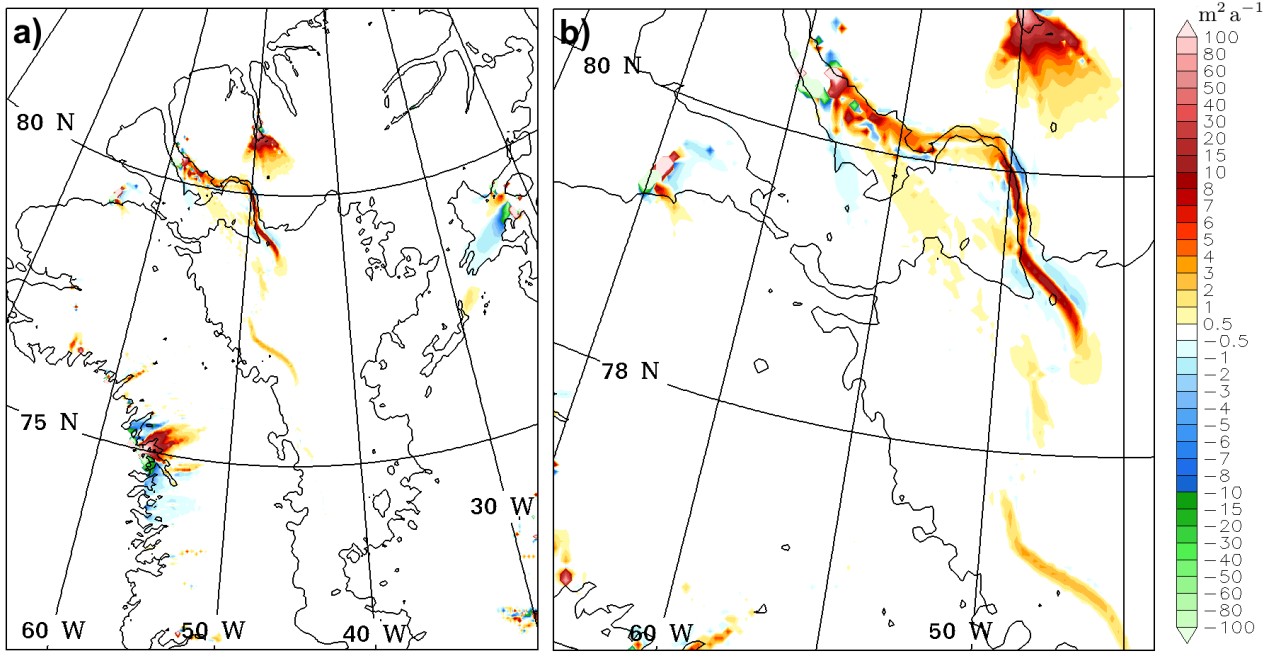

**Figure 6.** Surface ice velocity difference (Valley – Control) in metres per year for a) north Greenland and b) the Petermann catchment.

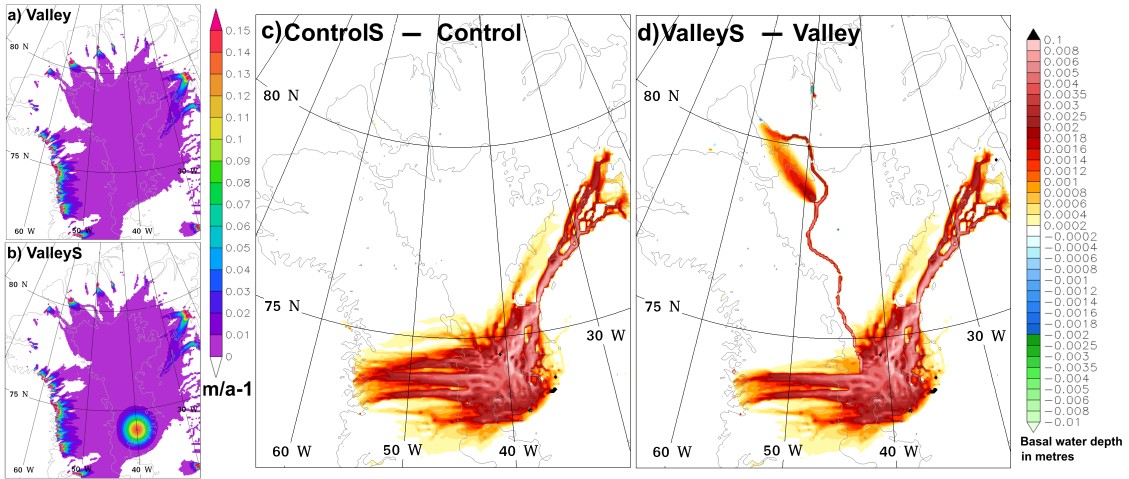

**Figure 7.** Basal melt rate $\mathrm{m\,a^{-1}}$ for a) Valley and b) ValleyS, and basal water depth differences (m) for c) ControlS - Control, and d) ValleyS - Valley.

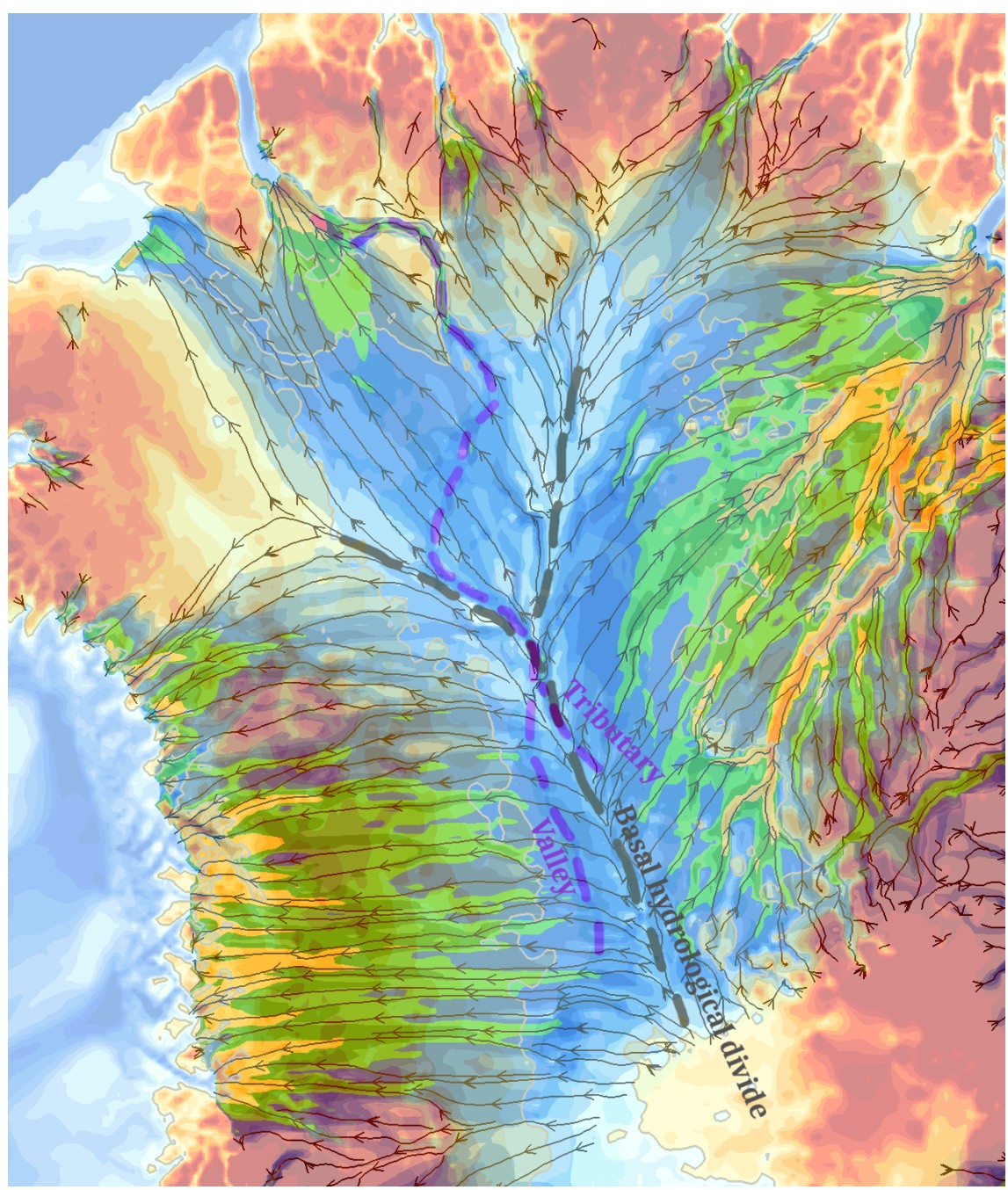

**Figure 8.** Schematic showing the location of the interior basal hydrological divide (grey dashes) simulated by SICOPOLIS and the path of the valley and the tributary (purple dashes). For guidance the background is the Control basal topography with the basal water flux from the sensitivity simulation that has a valley with fixed depth of $-100\,\text{m}$ (valley removal described in Appendix B) overlaid at 50% opacity.

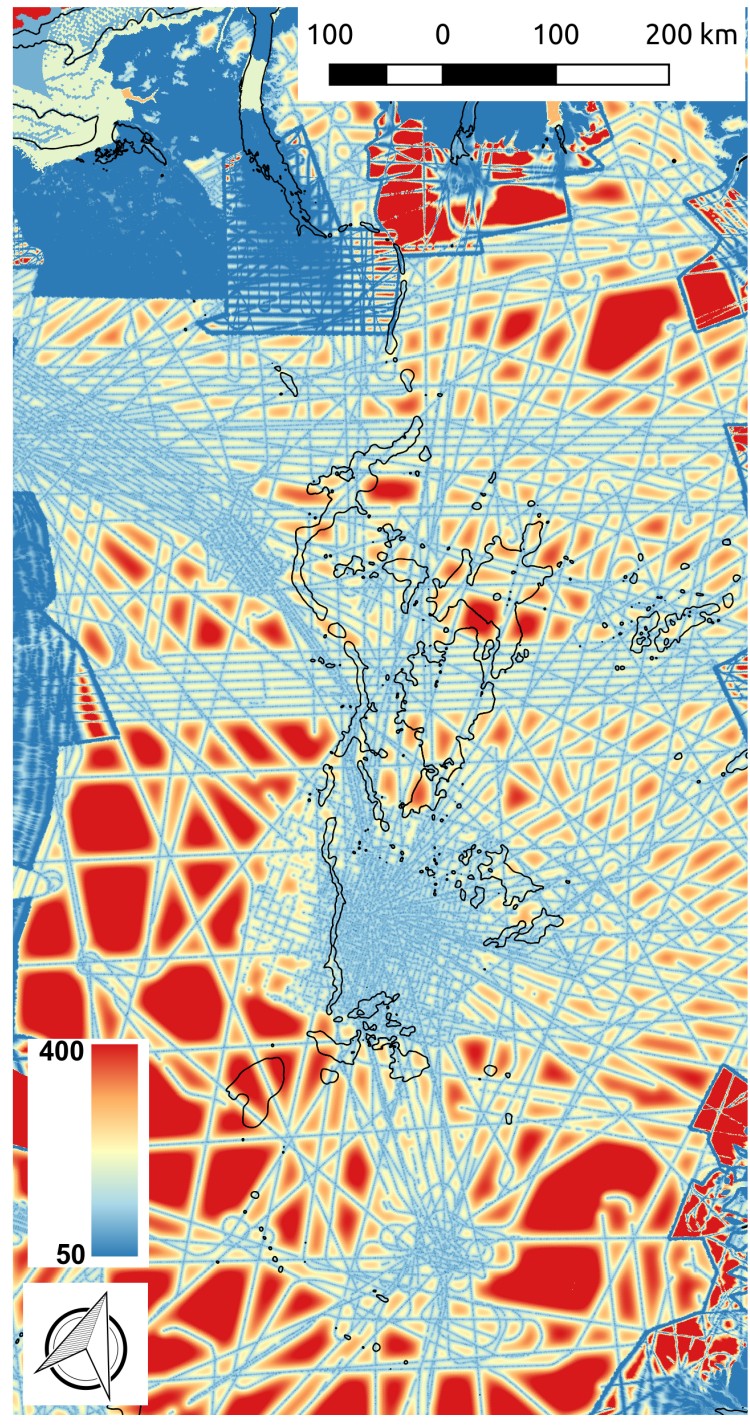

**Figure A1.** BedMachine basal topography error (Morlighem et al., 2017) in metres for the region from Petermann to Basin. The black contour indicates −200 metre elevation.

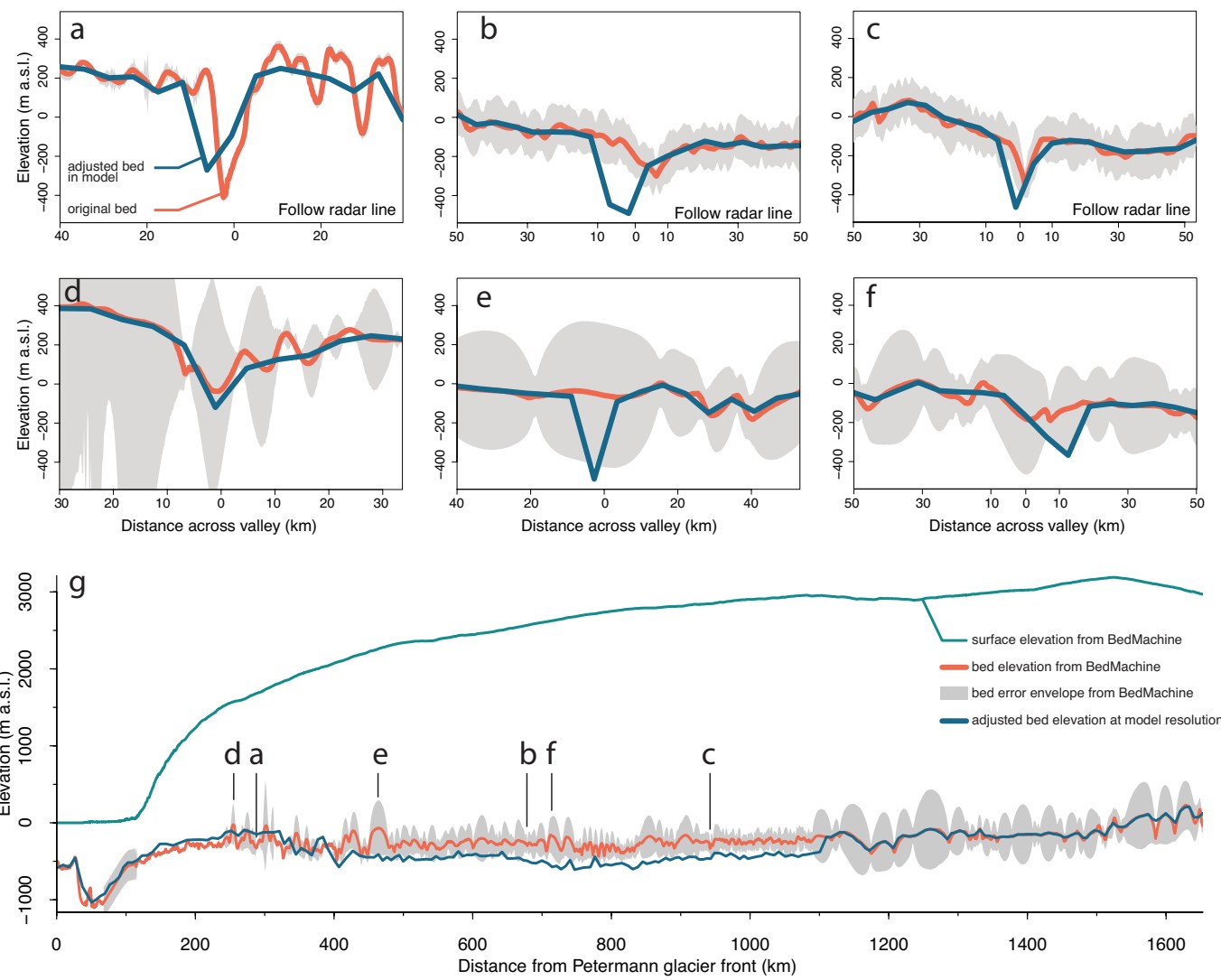

**Figure A2.** Across (a-f) and along (g) valley profiles from BedMachine v3 bed topography (Morlighem et al., 2017) and the adjusted bed elevation used in our model. The error envelope is derived from error estimates provided in BedMachine. Reduction in error depends on the proximity to radar data as shown in lines a-c that are parallel to flight lines or the use of mass conservation to derive bed topography which covers the region between 130 and 250 km in g).

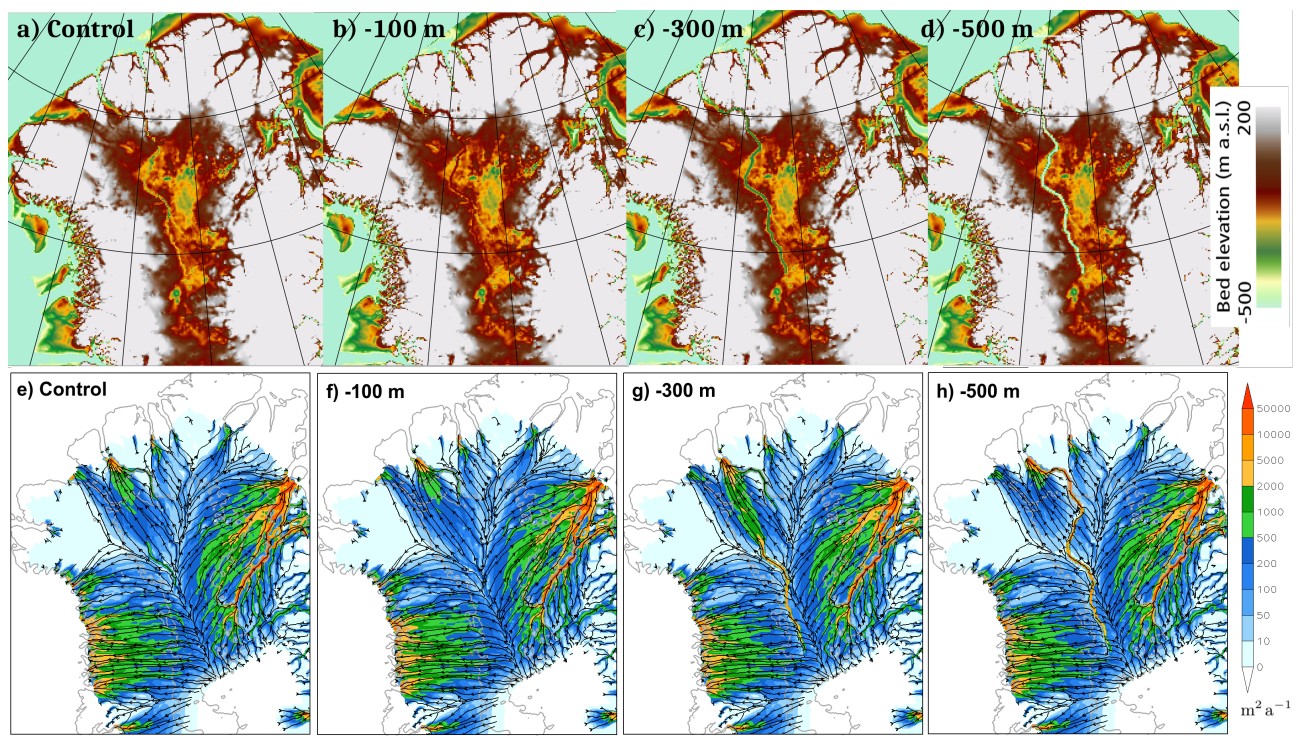

**Figure A3.** Sensitivity to valley depth tests. Bed topography (m) for a) Control, and for fixed valley base elevations (relative to sea level) of b) $-100\,\mathrm{m}$, c) $-300\,\mathrm{m}$, and d) $-500\,\mathrm{m}$. Basal water flux magnitude ($\mathrm{m^2\,a^{-1}}$ colours) and streamlines for north Greenland for e) Control, f) $-100\,\mathrm{m}$, g) $-300\,\mathrm{m}$, and h) $-500\,\mathrm{m}$.