# Peer review of "On the possibility of a 1000 km long subglacial river or wet sediment flow under the north Greenland ice sheet"

_The Cryosphere, 2019_

## Short Comment (SC1) · 29 Jul 2019

This is not a review of the paper nor have I gone through the whole manuscript in detail, but I have some specific comments. My first major issue is with the title. First, this is not a paper about the topic of the title at all. They present no evidence for the existence of a continuous "river" and that analysis was undertaken in a paper published six years ago [*Bamber et al.*, 2013]. The title of that paper answers the question about the existence of a continuous channel beneath the GrIS and we show hydraulic potentials that indicate likely drainage along the channel. Furthermore, we make it clear that the limited IPR coverage prevents direct observational evidence of a continuous channel but, given its origin, we conclude that it is most probably continuous. A large part of the introduction to this paper is, therefore, repeating previous inferences but in a way that suggest that they are new. I find that problematic.

Second, what do the authors mean by "river". To use this noun is incredibly misleading especially given the fact that, according to Fig 3b the modelled water depth appears to be ~0.5 mm! Water flow may be within an R channel (but not with those depths) or it may be via a thin film and it may or may not be continuous along the length of the channel. That is not a "river" in its conventional meaning.

A better title for this study would be "On the impact on subglacial water routing of a continuous subglacial channel from central to northern Greenland" because that is what this paper is actually about. Not conjectures on the existence of a "river" or not.

There appears to be either a problem with scale/labelling on Fig 3 or with the calculations themselves because the water depth in Fig 3b appears to have a maximum (plotted) depth of 1 cm. However the difference in water depth in the two cases (Fig 3d) is up to 10 cm, which is clearly impossible. Either there is a problem with the numeric or the plotting. In addition, I read the description of Fig 3 and I had difficulty in making sense of it. I could not discern, for example, the 3 quasi linear sections the authors refer to.

Bamber, J. L., Siegert, M. J., Griggs, J. A., Marshall, S. J., and Spada, G. (2013), Paleofluvial Mega-Canyon Beneath the Central Greenland Ice Sheet, *Science*, *341*(6149), 997-999.

---

## Referee Comment (RC1) · Anonymous Referee #1 · 23 Aug 2019

Review of Chamber et al

General Comments

This appears to be a poorly conceived study or, at least, a poorly conceived description of the study. It reproduces a key inference from a paper published six years ago and presents it as a new result or inference. As mentioned in the already posted comment, this study is actually about the influence of a continuous subglacial canyon on water flow and ice dynamics. While that is a topic that could be interest, almost all of the discussion presents flawed, hand-waving supposition that has little evidential basis and has little, if anything, to do with the model experiments.

The paper would benefit from a substantial rewrite and refocusing on the actual topic of the experiments conducted: what is the impact of a continuous canyon on the subglacial hydrology beneath the Greenland ice sheet? However, as the hydrology model used assumes thin film flow and no sources from surface melting are included, it is debatable to what extent the experiments can address this question with adequate confidence.

Specific Comments

1) As already mentioned in the posted comment, the title is not only extremely misleading it addresses a question that was tackled previously and this study provides no new evidence to answer it. The entire abstract needs to be rewritten focusing on the results of the model experiments and nothing else.

2) It is unclear to me why the authors have to introduce or replace existing terminology. They need to remove the word "river" throughout and replace with either conduit, R channel, or thin film as appropriate. In addition, why do they rename the canyon, "valley". Their explanation (p1, l23) is nonsensical.

3) The introduction is misleading and needs to be rewritten. Quoting directly from Bamber et al 2013 (B2013), they state "we present evidence from ice-penetrating radar data for a 750-km-long subglacial canyon in northern Greenland that is likely to have influenced basal water flow from the ice sheet interior to the margin"; "In all cases, above ~76° N and within the entire length of the Petermann catchment, the canyon exerts a control on basal water flow. For ~200 km, it provides an uninterrupted hydraulic pathway (Fig. 3 and fig. S3A) that ends at the terminus of Petermann Gletscher" and so on. I do not understand why the authors are proposing this as something entirely new.

4) P2 l1. The authors appear to be unaware that the bed topo in BedMachine v3 and that which was used in B2013 are essentially the same. The only difference is in the use of mass continuity near the margins where IPR coverage is poor. Any conclusions drawn in B2013 will be identical for BedMachine.

5) P4, l35. Fig 5 does not show slope, it shows surface elevation. The slope in the interior of ice sheets is small everywhere. This is not the same as "near flat" which is meaningless. Eyeballing the Fig it looks like the canyon follows the surface slope quite closely in the interior. This sentence is a good example of the hand-waving vagueness that pervades other parts of the paper.

6) P5, l5-6. I don't understand the logic of this statement. The canyon is not linear and doesn't follow the streamlines when it is not present (Fig 4a). This is nonsensical.

7) Section 4 Discussion. I found this section far too speculative, hand-waving and non-scientific. The first part is a qualitative overview of subglacial water flow theory. The arguments for why water may be present in the canyon are OK, in general although the discussion of enhanced GHF was a bit muddled and unclear. It could have all been stated in half the space as basal frictional heating, warmer ice at depth are all well established concepts. By page 7, l9 the discussion becomes too speculative. The authors appear to be unaware that B2013 examined hydraulic potentials for the isostatically compensated bed (Fig 3A) and discuss this in some detail. In addition, the authors do not explain how 300-800 m of bedrock erosion from subglacial water flow is possible during the Holocene. That would be a challenge even for rapid basal sliding over a soft bed, neither of which is the case here. At LGM, ice flow and hydraulic routing was different due to larger ice sheet cover.

8) P7, l27. Why?

9) P7, l30-32. This appears to demonstrate a complete lack of understanding of how the lithosphere responds to changes in ice loading. Why would there be a differential viscous response of the mantle across a < 10 km wide channel? In addition, how flat is the canyon and how flat is it after isostatic compensation? This is really incoherent.

10) Section 5. As for the abstract, title and Discussion, this section needs to be rewritten, focusing on the actual results and not wild and unsupported speculation.

---

## Author Comment (AC1) · 25 Aug 2019

Thanks a lot for your Short Comment. Please find below our response to the points and issues that you raise.

**1. Novelty**

"They present no evidence for the existence of a continuous "river" and that analysis was undertaken in a paper published six years ago [Bamber et al., 2013]. The title of that paper answers the question about the existence of a continuous channel beneath the GrIS and we show hydraulic potentials that indicate likely drainage along the channel. Furthermore, we make it clear that the limited IPR coverage prevents direct observational evidence of a continuous channel but, given its origin, we conclude that it is most probably continuous. A large part of the introduction to this paper is, therefore, repeating previous inferences but in a way that suggest that they are new. I find that problematic."

The novelty of the paper, that re-examines the canyon found by Bamber et al., 2013, is on:
- Demonstrating water flow along the entire length of the valley under current ice conditions based on an advanced ice-sheet model that also accounts for water production at the base of the ice sheet instead of mapping potential subglacial water pathways based on Shreve, 1972's hydraulic potential theory.
- Demonstrating that our current picture of the Greenland ice-sheet subglacial hydrology underestimates the impacts from interpolation error artifacts and that by correcting these errors, large changes to the basal water distribution can occur.

The evidence from our simulations show an uninterrupted subglacial water pathway interpreted as a subglacial river system from the Interior of the Greenland Ice Sheet to Petermann glacier once the valley is opened. This was not done by Bamber et al. (2013) who presented subglacial hydraulic potential flow paths based on ice-sheet bed data that include the numerous rises in the valley caused by kriging interpolation.

Furthermore, we present other pieces of evidence and supportive of a currently active subglacial river system associated with the relation to ice thickness, basal hydrological basins, and the flatness of the valley base. We will reword the relevant paragraph in the introduction and better cite Bamber et al., 2013 with respect to their discussion of the possible continuity of the valley and the effects of limited IPR coverage.

**2. Title**

"Second, what do the authors mean by "river". To use this noun is incredibly misleading especially given the fact that, according to Fig 3b the modelled water depth appears to be ~0.5 mm! Water flow may be within an R channel (but not with those depths) or it may be via a thin film and it may or may not be continuous along the length of the channel. That is not a "river" in its conventional meaning.
A better title for this study would be "On the impact on subglacial water routing of a continuous subglacial channel from central to northern Greenland" because that is what this paper is actually about. Not conjectures on the existence of a "river" or not."

We would prefer to keep the original title after considering the following points:
- The article is about the "possibility of a long subglacial river" and as such presents evidence for such a system while recognizing that the huge void in data precludes a conclusive result.

- The word "river" is appropriate when defined as "subglacial river" which has different properties to a river over open land. In addition, many "rivers" on Earth do not flow at all times, or flow at certain times only along certain sections.

In terms of the very shallow simulated water depth in the valley, as mentioned in the manuscript, our basal hydrological model cannot simulate rivers so it is impossible to attain any appreciable depth unless the flood-fill algorithm is activated.

**3. Water sheet depth in Figure 3**

"There appears to be either a problem with scale/labelling on Fig 3 or with the calculations themselves because the water depth in Fig 3b appears to have a maximum (plotted) depth of 1 cm. However the difference in water depth in the two cases (Fig 3d) is up to 10 cm, which is clearly impossible. Either there is a problem with the numeric or the plotting. In addition, I read the description of Fig 3 and I had difficulty in making sense of it. I could not discern, for example, the 3 quasi linear sections the authors refer to."

With regards to the scale labeling on Figure 3, there is no error in the plot. For most of the values in the difference plot the depth change is less than 0.008 metres. However, the top of the colour bar indicates black regions of over 0.1 metres difference (increase). The black regions (of which there are only 4) represent positions where the flood fill algorithm has filled an area leading to a depth increase of over 10 cm. While our typical water depths are the order of millimetres, the fill algorithm can produce depths up to a threshold of 10 metres. We will make this clearer in the next version of the manuscript.

Bamber, J. L., Siegert, M. J., Griggs, J. A., Marshall, S. J., and Spada, G. (2013), Paleofluvial Mega-Canyon Beneath the Central Greenland Ice Sheet, Science, 341(6149), 997-999

---

## Referee Comment (RC2) · Andy Aschwanden (Referee) · 27 Aug 2019

Review of Chambers et al

Sorry for the delay, I was doing field work last week. I wrote parts of my review before other comments were posted, thus some of my comments may be redundant.

Best wishes,

Andy Aschwanden, University of Alaska Fairbanks

Review of "On the possibility of a long subglacial river under the north Greenland ice sheet"

I was quite excited when I saw this topic presented at AGU 2018, and happily accepted

the review. The manuscript is generally well written and I find the idea of assessing the impact of the canyon on ice flow a worthwhile endeavor. Unfortunately in its currently form the study is poorly designed and a missed opportunity.

First (as already pointed out by Jonathan Bamber), the title is misleading because Bamber et al (2013) [B13a] already demonstrated the existence of a long subglacial channel. I suggest to change the title to something that more closely reflects the core of this manuscript. E.g. along the lines of "On the influence of a long subglacial channel on basal hydrology and ice flow".

The hydrological and ice flow modeling, which is the most novel aspect of this manuscript, is ill-conceived. The authors compare a simulation "Valley" to a "Control" simulation which to me has limited value. Why not compare the simulation to the readily-available surface velocity measurements, as one would hope that better physics leads to a closer agreement with observations? In Figure 6 the differences between "Valley" and "Control" exceed 10 m yr-1. Looking at observation of surface speeds, I find no clear signature of a channel, contrary to what is expected. I wonder if this means that the sliding model is too sensitive to changes in basal water. Furthermore, with a channel width of 5-10km, it is not clear to me how well a model resolution of 5km is able to capture the dynamics that the authors are interested in.

Right now the manuscript consists of a lot of material that was already discussed by B13a, an interesting but not well thought out modeling part, and a very speculative discussion on geothermal flux. While the manuscript has potential, I cannot recommend it for publication in its current form; a conclusion I did not reach lightheartedly. Maybe these suggestions can help to rewrite the manuscript, with focus on 1. what is novel compared B13a 2. the ice flow modeling and how the inclusion of the channel improves agreement with the observed flow structure.

Detailed comments:

P2, L 1: Please note that in the interior of the ice sheet, BedMachine is based on kriging, mass conservation was only used near the coast. It is thus more or less equivalent to bed map of Bamber et al (2013) [B13b].

P2, L 11: change "when you consider", this is too colloquial

P2, L23: Maybe I don't understand the initialization procedure correctly, but why is 1990 the target date? P2 L 30 that corrections are made to bring the simulated ice thickness in agreement with observations. But to the best of my knowledge there is no 1990m DEM of Greenland available that can be used as a target?

P5, L5: "extremely gently" sounds awkward. Maybe just "gently" or "very gently"?

P5, L 25: "where you get" is too colloquial.

Figures: To increase readability, I recommend using the same color scales for bedrock elevation in all figures. Currently there a 3 color scales (Fig 1, 2, 5).

Fig 4: A close up of the NEEM zone would be helpful, the flow lines are hard to distinguish here.

References:

[B13a] Bamber, J. L. et al. (2013) 'Paleofluvial Mega-Canyon Beneath the Central Greenland Ice Sheet', Science, 341(6149), pp. 997–999. doi: 10.1126/science.1239794.

[B13b] ]Bamber, J. L. et al. (2013) 'A new bed elevation dataset for Greenland', The Cryosphere, 7(2), pp. 499–510. doi: 10.5194/tc-7-499-2013.

---

## Author Comment (AC2) · 25 Sep 2019

**Response to Reviewer 1**

Thank you very much for taking the time to review our paper. Please find below our responses to your comments. *Reviewer statements are in italics.*

*"This appears to be a poorly conceived study or, at least, a poorly conceived description of the study. It reproduces a key inference from a paper published six years ago and presents it as a new result or inference."*

This study is the first demonstration of how the opening of a long valley in the bedrock can impact the subglacial hydrology and ice sheet sliding in northern Greenland. The results show that even small adjustments in the bed topography to include probable features can have consequences that could affect future simulations of the ice sheet. Bamber et al. 2013 (B2013) did not demonstrate that an uninterrupted water pathway was possible from Interior to Petermann. Their figures show water pathways along numerous independent sections of the valley. They infer that the valley is probably continuous but do not present scientific evidence of the potential effects the continuity of the feature could have on the basal hydrology or ice sheet sliding. Our results are new and we hope will be useful for progressing forward in our understand of subglacial valley features. Areas of the introduction and discussion can be reworded to reference the inferences made in B2013 which we hope will address this concern.

*"As mentioned in the already posted comment, this study is actually about the influence of a continuous subglacial canyon on water flow and ice dynamics. While that is a topic that could be interest, almost all of the discussion presents flawed, hand-waving supposition that has little evidential basis and has little, if anything, to do with the model experiments."*

The discussion outlines the potential significance of the valley based on the model results and current form of the ice sheet and known bed topography. We are combining the techniques of model analysis with observational data and discuss the implications in the context of past research on subglacial hydrology. Responses to specific comments raised concerning the discussion are below.

*"The paper would benefit from a substantial rewrite and refocusing on the actual topic of the experiments conducted: what is the impact of a continuous canyon on the subglacial hydrology beneath the Greenland ice sheet? However, as the hydrology model used assumes thin film flow and no sources from surface melting are included, it is debatable to what extent the experiments can address this question with adequate confidence."*

We are making changes to the manuscript to firm up our investigation, including making discharge estimates. The thin-film model is used as a guide for where subglacial water is moving and collecting under the ice sheet.

Contributions from surface melting to basal water are currently impossible to accurately assess but would only serve to enhance the probability of a subglacial river being present within the valley. Given the path of the valley mostly within the ice sheet interior, it may be difficult for surface melt water to reach the base, at least under present conditions. It's an interesting and important issue and one that can be mentioned in the discussion.

*Specific Comments*
*1) As already mentioned in the posted comment, the title is not only extremely misleading it addresses a question that was tackled previously and this study provides no new evidence to answer it. The entire abstract needs to be rewritten focusing on the results of the model*

*experiments and nothing else.*

The finding of an uninterrupted water pathway along the valley from Interior to Petermann was not demonstrated in previous work. B2013 Figure 3 demonstrates the effects that errors in bedrock data have on water pathways that consequently fail to follow the valley route. We demonstrate for the first time the potential consequences of accounting for these errors by opening one single valley. We think it is better to use model results in conjunction with observations for this type of study rather than solely focus on the model results.

*2) It is unclear to me why the authors have to introduce or replace existing terminology. They need to remove the word "river" throughout and replace with either conduit, R channel, or thin film as appropriate. In addition, why do they rename the canyon, "valley". Their explanation (p1, l23) is nonsensical.*

The term "subglacial river" is used here to cover a variety of possible subglacial river forms such as within R-channels, canals, Nye channels, or braided rivers. We will add our definition of this to the introduction.

We use the term "valley" because in areas of the interior the feature does not appear to take the form of a canyon as for example defined as "a deep valley with steep sides". In regions the valley does not have steep sides, so we think the broader term "valley" is more appropriate as it better accounts for a wider range of forms. In work prior to B2013 the feature was referred to a "bedrock trough" or "trench" by Ekholm et al. (1998) , and a "channel" or "subglacial valley" by van der Veen et al. (2007). In addition, the term "tunnel valley" has been widely used for subglacial water incised channels. The terminology can be changed if the editors, reviewers and co-authors think that "canyon" is a more appropriate term to use for the feature as a whole.

*3) The introduction is misleading and needs to be rewritten. Quoting directly from Bamber et al 2013 (B2013), they state "we present evidence from ice-penetrating radar data for a 750-km-long subglacial canyon in northern Greenland that is likely to have influenced basal water flow from the ice sheet interior to the margin"; "In all cases, above ~76° N and within the entire length of the Petermann catchment, the canyon exerts a control on basal water flow. For ~200 km, it provides an uninterrupted hydraulic pathway (Fig. 3 and fig. S3A) that ends at the terminus of Petermann Gletscher" and so on. I do not understand why the authors are proposing this as something entirely new.*

These are not the same as the results are presented in our paper. B2013 present evidence in their Figure 3 of water flow along some sections of the valley but not along the whole length in any of the scenarios. With present day ice sheet cover they find an uninterrupted water pathway for only the northernmost section ~200 km in Petermann catchment as stated. This demonstrates the strong forcing for water flow along this section when ice is present because this occurs even in these cases that contain large and unrealistic artificial blocks in the valley. Nonetheless even in these cases, it is worth noting that water is diverted away from the valley just prior to entering the Petermann Glacier basin due to a large artificial block in the bed topography there. This error also occurs in the results of Chu (2017) Figure 4.6c. Further inland the situation is more serious with water flowing in and out of the valley in numerous locations in all of the cases presented in B2013. The only result in our paper that is similar to the distribution in B2013 Figure 3c is our Figure 4a which presents our control case. This is because it also contains these unrealistic blocks in the valley.

*4) P2 l1. The authors appear to be unaware that the bed topo in BedMachine v3 and that which was used in B2013 are essentially the same. The only difference is in the use of mass continuity near the margins where IPR coverage is poor. Any conclusions drawn in B2013 will be identical*

*for BedMachine.*

Our tests are focused on the effects of the removal of blocks in the valley in BedMachine v3. We are not presenting any results or discussion concerning differences between BedMachine v3 and the data used in B2013.

*5) P4, l35. Fig 5 does not show slope, it shows surface elevation. The slope in the interior of ice sheets is small everywhere. This is not the same as "near flat" which is meaningless. Eyeballing the Fig it looks like the canyon follows the surface slope quite closely in the interior. This sentence is a good example of the hand-waving vagueness that pervades other parts of the paper.*

We can change the terminology from "near flat" to a more appropriate term. The valley progresses down under a very gentle ice surface slope that would not oppose subglacial water flow in its base.

*6) P5, l5-6. I don't understand the logic of this statement. The canyon is not linear and doesn't follow the streamlines when it is not present (Fig 4a). This is nonsensical*

In this section we are trying to establish whether the current form and path of the valley is consistent with what could be expected for a subglacial river under the current ice sheet configuration. While the valley appears to intersect the east and west hydrological basins in the interior it does not follow the streamlines, particularly as it crosses NEEM zone. A perfectly aligned valley with present day streamlines is probably not to be expected given the different extents of the ice sheet in the past. B2013 Figure 3b is a good indicator of this as the basal hydrological divide is shifted to the west under LGM conditions and the valley still follows a path not far off the basal hydrological divide. We can reword and add some sentences discussing this issue and reference B2013 Figure 3b.

*7) Section 4 Discussion. I found this section far too speculative, hand-waving and non-scientific. The first part is a qualitative overview of subglacial water flow theory. The arguments for why water may be present in the canyon are OK, in general although the discussion of enhanced GHF was a bit muddled and unclear. It could have all been stated in half the space as basal frictional heating, warmer ice at depth are all well established concepts.*

This section on water flow and geothermal heat flux can be made more concise. The overview on subglacial water theory is necessary to address the key issue of the paper on whether water flow within the valley is likely to be in the form of a subglacial river.

*By page 7, l9 the discussion becomes too speculative.*

Given the possible large extension of the valley catchment to Basin we think it is important to discuss this in this section. We try to make it clear in the discussion where we are uncertain.

*The authors appear to be unaware that B2013 examined hydraulic potentials for the isostatically compensated bed (Fig 3A) and discuss this in some detail. In addition, the authors do not explain how 300-800 m of bedrock erosion from subglacial water flow is possible during the Holocene. That would be a challenge even for rapid basal sliding over a soft bed, neither of which is the case here. At LGM, ice flow and hydraulic routing was different due to larger ice sheet cover.*

The overall erosive history of this valley is likely to be complicated and difficult or impossible to determine. Our results are focused on whether the present day form, in terms of it's path (in relation to ice surface slope) and along valley base slope, is compatible with a subglacial river under a thick

ice sheet. At LGM the hydraulic routing according to B2013 Fig 3b still indicates the valley roughly following the hydrological divide in the interior, suggesting that conditions for along valley water routing there could have been still been favourable then. The possibility remains that the present day valley form developed as a consequence of erosion under some or all of the following 1) under current conditions, 2) under LGM conditions, 3) during ice sheet retreat, 4) under reduced ice sheet cover, and 5) under ice free conditions. We simply do not have enough information. Tunnel valleys provide one demonstration of how subglacial water erosion can erode into hundreds of metres of bedrock. Given that the source is close to a proposed geothermal warm spot, past episodic subglacial down-valley discharges are a possibility. Much smaller amounts of erosion and deposition would be needed to maintain a base slope favourable for water routing, as is typical of rivers in general.

*P7, l27. Why?*

The concept here is that it seems unlikely that, in the absence of subglacial river erosion, a river formed when there was no ice sheet would take a form that would end up also being favourable for subglacial water flow under thick ice. In this sense the substantial changes to the form of the bedrock caused by the build up of km thick ice should make a roughly level paleo river valley capable of routing subglacial water all the way out from the interior, unlikely. Having said that the valley base does not appear to be perfectly level by any means, and does not follow the hydrological divide perfectly so we will need to clarify these points in the discussion.

*9) P7, l30-32. This appears to demonstrate a complete lack of understanding of how the lithosphere responds to changes in ice loading. Why would there be a differential viscous response of the mantle across a < 10 km wide channel? In addition, how flat is the canyon and how flat is it after isostatic compensation? This is really incoherent.*

We are referring to the along-valley differential viscous response, not the across valley one. A present day relatively level valley base over such a long distance is consistent with subglacial erosive and depositional water activity but does not prove it. The isostatic correction in B2013 Fig. S5 implies that on an ice-free Greenland the valley would be 600 m higher than present in the interior progressing to 200 metres higher near the coast. Today the valley base appears to be consistently between -300 and -500 metres with no clear trend along-valley.

*10) Section 5. As for the abstract, title and Discussion, this section needs to be rewritten, focusing on the actual results and not wild and unsupported speculation.*

We hope that our responses address some of your concerns and are ready to make changes where issues have been raised.

We would prefer to keep the original title after considering the following points:
- The article is about the "possibility of a long subglacial river" and as such presents evidence for such a system while recognizing that the huge void in data precludes a conclusive result.
- The word "river" is appropriate when defined as "subglacial river" which has different properties to a river over open land. In addition, many "rivers" on Earth do not flow at all times, or flow at certain times only along certain sections.

**References**

[B2013] Bamber, J. L., Siegert, M. J., Griggs, J. A., Marshall, S. J., and Spada, G. (2013), Paleofluvial Mega-Canyon Beneath the Central Greenland Ice Sheet, Science, 341(6149), 997-999

Ekholm, S., Keller, K., Bamber, J. L., and Gogineni, S. P.: Unusual surface morphology from digital elevation models of the Greenland ice sheet, Geophys. Res. Lett., 25, 3623–3626, https://doi.org/10.1029/98GL02589, 1998.

van der Veen, C. J., Leftwich, T., von Frese, R., Csatho, B. M., and Li, J.: Subglacial topography and geothermal heat flux: Potential interactions with drainage of the Greenland ice sheet, Geophys. Res. Lett., 34, L12 501, https://doi.org/10.1029/2007GL030046, 2007.

Wing Yin Chu 2017 Variability of Subglacial Drainage Across the Greenland Ice Sheet: A Joint Model/Radar Study. PhD thesis, Columbia University

---

## Author Comment (AC3) · 25 Sep 2019

**Response to Reviewer 2: Andy Aschwanden**

Thank you very much for taking the time to review our paper. We are sorry to hear that you are not recommending this study for publication and appreciate your careful consideration in coming to that decision. Nonetheless we would like to address the issues that you raised. *(Reviewer statements are in italics).*

*"First (as already pointed out by Jonathan Bamber), the title is misleading because Bamber et al (2013) [B13a] already demonstrated the existence of a long subglacial channel. I suggest to change the title to something that more closely reflects the core of this manuscript. E.g. along the lines of "On the influence of a long subglacial channel on basal hydrology and ice flow"."*

The title has been chosen because the focus of our investigation has been on the possibility of a currently active long subglacial river system along the valley route. We would prefer to keep the original title after considering the following points:
- The article is about the "possibility of a long subglacial river" and as such presents evidence for such a system while recognizing that the huge void in data precludes a conclusive result.
- The word "river" is appropriate when defined as "subglacial river" which has different properties to a river over open land. In addition, many "rivers" on Earth do not flow at all times, or flow at certain times only along certain sections." We are going to add this definition to the introduction to be clear and we are considering adding an additional appendix that discusses potential discharge estimates.
- We are also going to add a section to the discussion that details reasons why there may not be a subglacial river within the valley

Regarding the originality of the study, B13a did not demonstrate that an uninterrupted water pathway was possible under present day conditions from Interior to Petermann in the evidence they presented. Their figures show water pathways along numerous independent sections of the valley. They infer that the valley is probably continuous but do not present scientific evidence of the potential effects the continuity of the feature could have on the basal hydrology or ice sheet sliding. Therefore our results are original. We will add references to B13a beyond what is already included to make it clear what their results showed, how they differ from ours, and how our results relate to inferences made in B13a.

*"The hydrological and ice flow modeling, which is the most novel aspect of this manuscript, is ill-conceived. The authors compare a simulation "Valley" to a "Control" simulation which to me has limited value. Why not compare the simulation to the readily-available surface velocity measurements, as one would hope that better physics leads to a closer agreement with observations?"*

Thanks for the suggestion to compare with observations. Our goal in relation to ice sheet sliding is to provide an assessment of what impact an open valley introduced into a simulation can have. To do this it makes sense to compare two cases that are identical aside from the topographic modification, as we have done with "Valley" and "Control". The results therefore only show the consequences of the topographic change and so reveal the influence it has. For assessing whether the introduction of a valley improves the simulation with respect to observations it would be valuable to make the comparison you suggest, however this has not been our aim up to this point.

*"In Figure 6 the differences between "Valley" and "Control" exceed 10 m yr-1. Looking at observation of surface speeds, I find no clear signature of a channel, contrary to what is expected. I wonder if this means that the sliding model is too sensitive to changes in basal water."*

[Figure]

*Figure 1: Surface ice velocity difference (Valley – Control) in metres per year for a) north Greenland and b) the Petermann catchment. Grey box indicates the region used in Figure 2.*

The figure above highlights the relevant areas. There are two regions of increased sliding in the Petermann catchment. One associated with the Petermann Ice Stream region and another narrower one associated with the valley.

We did the simulations with the shallow-ice approximation (SIA), which produces a too localized response of the ice flow to changed basal sliding. However, we are planning to re-do the simulations with the hybrid shallow-ice--shelfy-stream scheme that captures the dynamics of fast-flowing grounded ice more realistically. This should lead to a more distributed response of the surface velocity, thus a less pronounced signature at the surface.

Other possible explanations for the lack of a 10 m a$^{-1}$ speed up directly over the valley in the observed surface velocities include:
1) There is no subglacial river, or significant subglacial water within this section of the valley, i.e. it is mostly frozen to the bed.
2) Since the model relies on a thin film basal hydrology it fails to model any type of subglacial river. Consequently it fails to channelize the water, potentially causing an unrealistic distribution of water in a wide film that has a greater influence on sliding.
3) As you suggest, the sliding model could be too sensitive to changes in basal water.

This issue will be noted in the paper.

The valley in this region is noticeable in the observed surface velocity field when you focus on speeds between 10 to 40 m a$^{-1}$ (Figure 2 below). However this is not like what we see in the model and may simply be due to the topography rather than basal water.

[Figure]

*Figure 2: Observed surface velocity in the coastal valley section.*

For the second area in the vicinity of the Petermann Ice Stream, differences exceeding 10 m yr-1 only occur in areas where the ice flows pretty fast, so that the differences amount to only a few percent. This does not produce a notable signature, and it is too small to allow a reliable judgement about whether "Valley" or "Control" matches observed surface velocities better.

*"Furthermore, with a channel width of 5-10km, it is not clear to me how well a model resolution of 5km is able to capture the dynamics that the authors are interested in."*

This is a significant model issue and it is one that important to highlight. Unfortunately we are currently unable to simulate the Greenland Ice Sheet at higher resolutions than 5 km and this restricts our ability to correctly reproduce basal topographic features. We can only present the results we have and detail potential error sources. We will add further mention of this issue. Judging by the basal water fluxes we present, we have successfully removed the blocks along the valley route, which was our goal. From a modeling perspective, this is a first step to demonstrate the potential extent of the impacts.

*"Right now the manuscript consists of a lot of material that was already discussed by B13a, an interesting but not well thought out modeling part, and a very speculative discussion on geothermal flux. While the manuscript has potential, I cannot recommend it for publication in its current form; a conclusion I did not reach lightheartedly. Maybe these suggestions can help to rewrite the manuscript, with focus on 1. what is novel compared B13a 2. the ice flow modeling and how the inclusion of the channel improves agreement with the observed flow structure."*

We present new scientific evidence that was not presented by B2013a and will reference B2013a further where appropriate. We hope that the points above help to address the issues presented on the modeling aspects. We can improve the discussion on geothermal heat flux if we proceed further with the paper.

*"Detailed comments:*
*P2, L 1: Please note that in the interior of the ice sheet, BedMachine is based on kriging, mass conservation was only used near the coast. It is thus more or less equivalent to bed map of Bamber et al (2013) [B13b]."*

This is noted. Our study is focused on adjustments made to open the valley rather than any minor changes in the data since B13b. For reference the additional flight lines used since 2013 are shown in the figure below.

[Figure]

*"P2, L 11: change "when you consider", this is too colloquial"*

Thanks. We will make this change.

*"P2, L23: Maybe I don't understand the initialization procedure correctly, but why is 1990 the target date? P2 L 30 that corrections are made to bring the simulated ice thickness in agreement with observations. But to the best of my knowledge there is no 1990m DEM of Greenland available that can be used as a target?"*

Our SICOPOLIS model setup is based on that used for the Ice Sheet Model Intercomparison Project for CMIP6 (ISMIP6). The reason why we chose 1990 as the initialization date for ISMIP6 is that the Greenland ice sheet was approximately in balance around that time, while its mass balance has become more and more negative since then. You are correct that there is no DEM for 1990. The nominal date for BedMachine Greenland v3, from which we also took the surface DEM, is 2007 <https://sites.uci.edu/morlighem/dataproducts/bedmachine-greenland/>.
However, the difference between the overall topography 1990 and 2007 is very small, so that it doesn't really matter. More important is that the climate forcing for the spin-up run ends in 1990 and produces an ice sheet that is very much in balance.

*"P5, L5: "extremely gently" sounds awkward. Maybe just "gently" or "very gently"?*

*P5, L 25: "where you get" is too colloquial."*

We will make these changes.

*"Figures: To increase readability, I recommend using the same color scales for bedrock elevation in all figures. Currently there a 3 color scales (Fig 1, 2, 5).*

*Fig 4: A close up of the NEEM zone would be helpful, the flow lines are hard to distinguish here."*

We can make these changes if we proceed further with publication. Thanks for the suggestions.

**References**

[B13a] Bamber, J. L. et al. (2013) 'Paleofluvial Mega-Canyon Beneath the Central Greenland Ice Sheet', Science, 341(6149), pp. 997–999. doi: 10.1126/science.1239794.

[B13b] ]Bamber, J. L. et al. (2013) 'A new bed elevation dataset for Greenland', The Cryosphere, 7(2), pp. 499–510. doi: 10.5194/tc-7-499-2013.

---

## Author Response (AR1)

**Response to Reviewers**

**Table of Contents**

| Response to Reviewer 1                  | 1   |
|-----------------------------------------|-----|
| Response to Reviewer 2: Andy Aschwanden | 5   |
| References                              | 8   |
| Marked up manuscript                    | . 9 |
|                                         | 9   |

**Response to Reviewer 1**

Thank you very much for taking the time to review our paper. Please find below our responses to your comments. *Reviewer statements are in italics*.

"This appears to be a poorly conceived study or, at least, a poorly conceived description of the study. It reproduces a key inference from a paper published six years ago and presents it as a new result or inference."

This study is the first demonstration of how the opening of a long valley in the bedrock can impact the subglacial hydrology and ice sheet sliding in northern Greenland. Bamber et al. 2013 (B2013a) did not demonstrate that an uninterrupted water pathway was possible from Interior to Petermann. Their figures show water pathways along numerous independent sections of the valley. They infer that the valley is probably continuous but do not present scientific evidence of the potential effects the continuity of the feature could have on the basal hydrology or ice sheet sliding. Our results are new and we hope will be useful for progressing forward in our understand of subglacial valley features. Areas of the introduction and discussion have been reworded to reference the inferences made in B2013a and additional results are presented to further the originality of this study which we hope will address these concerns.

"As mentioned in the already posted comment, this study is actually about the influence of a continuous subglacial canyon on water flow and ice dynamics. While that is a topic that could be interest, almost all of the discussion presents flawed, hand-waving supposition that has little evidential basis and has little, if anything, to do with the model experiments."

The discussion outlines the potential significance of the valley based on the model results and current form of the ice sheet and known bed topography. We are combining the techniques of model analysis with observational data and discuss the implications in the context of past research on subglacial hydrology. Responses to specific comments raised concerning the discussion are below.

"The paper would benefit from a substantial rewrite and refocusing on the actual topic of the experiments conducted: what is the impact of a continuous canyon on the subglacial hydrology beneath the Greenland ice sheet? However, as the hydrology model used assumes thin film flow and no sources from surface melting are included, it is debatable to what extent the experiments can address this question with adequate confidence."

We have made changes to the manuscript to firm up our investigation, including considering discharge estimates. The thin-film model is used as a guide for where subglacial water is moving and collecting under the ice sheet. The following statement has been changed:

"thin-film model is used as a guide for where subglacial water is moving and collecting under the ice sheet. Where the film is thickest along an uninterrupted path to an ocean entry is considered to be a sign of a basal environment with an increased likelihood of consisting of some form of subglacial river system."

Contributions from surface melting to basal water are currently impossible to accurately assess but would only serve to enhance the probability of a subglacial river being present within the valley. Given the path of the valley mostly within the ice sheet interior, it may be difficult for surface melt water to reach the base, at least under present conditions. It's an interesting and important issue and is briefly mentioned in the discussion:

"As the valley approaches closer to Petermann Fjord, contributions to the discharge from summer surface melting become more likely and have the potential to overwhelm any discharge from interior basal melting."

**Specific Comments**

1) As already mentioned in the posted comment, the title is not only extremely misleading it addresses a question that was tackled previously and this study provides no new evidence to answer it. The entire abstract needs to be rewritten focusing on the results of the model experiments and nothing else.

The finding of an uninterrupted water pathway along the valley from Interior to Petermann was not demonstrated in previous work. B2013a Figure 3 demonstrates the effects that errors in bedrock data have on water pathways that consequently fail to follow the valley route. We demonstrate for the first time the potential consequences of accounting for these errors by opening one single valley. We think it is better to use model results in conjunction with observations for this type of study rather than solely focus on the model results.

2) It is unclear to me why the authors have to introduce or replace existing terminology. They need to remove the word "river" throughout and replace with either conduit, R channel, or thin film as appropriate. In addition, why do they rename the canyon, "valley". Their explanation (p1, l23) is nonsensical.

The term "subglacial river" is used here to cover a variety of possible subglacial river forms such as within R-channels, canals, Nye channels, or braided rivers. We will add our definition of this to the introduction.

We use the term "valley" because in areas of the interior the feature does not appear to take the form of a canyon as for example defined as "a deep valley with steep sides". In regions the valley does not have steep sides, so we think the broader term "valley" is more appropriate as it better accounts for a wider range of forms. In work prior to B2013a the feature was referred to a "bedrock trough" or "trench" by Ekholm et al. (1998) , and a "channel" or "subglacial valley" by van der Veen et al. (2007). In addition, the term "tunnel valley" has been widely used for subglacial water incised channels. The terminology can be changed if the editors, reviewers and co-authors think that "canyon" is a more appropriate term to use for the feature as a whole. The following statement has been edited:

"Since this "trench" (Ekholm et al., 1998), "subglacial valley" (van der Veen et al., 2007) or "canyon" (Bamber et al., 2013) takes a variety of cross-sectional forms along its length, in this article we will simply refer to it as a in the broadest term as a subglacial "valley"."

3) The introduction is misleading and needs to be rewritten. Quoting directly from Bamber et al

2013 (B2013a), they state "we present evidence from ice-penetrating radar data for a 750-kmlong subglacial canyon in northern Greenland that is likely to have influenced basal water flow from the ice sheet interior to the margin"; "In all cases, above ~76° N and within the entire length of the Petermann catchment, the canyon exerts a control on basal water flow. For ~200 km, it provides an uninterrupted hydraulic pathway (Fig. 3 and fig. S3A) that ends at the terminus of Petermann Gletscher" and so on. I do not understand why the authors are proposing this as something entirely new.

These are not the same as the results presented in our paper. B2013a present evidence in their Figure 3 of water flow along some sections of the valley but not along the whole length in any of the scenarios. With present day ice sheet cover they find an uninterrupted water pathway for only the northernmost section ~200 km in Petermann catchment as stated. This demonstrates the strong forcing for water flow along this section when ice is present because this occurs even in these cases that contain large and unrealistic artificial blocks in the valley. Nonetheless even in these cases, it is worth noting that water is diverted away from the valley just prior to entering the Petermann Glacier basin due to a large artificial block in the bed topography there. This error also occurs in the results of Chu (2017) Figure 4.6c. Further inland the situation is more serious with water flowing in and out of the valley in numerous locations in all of the cases presented in B2013a. The only result in our paper that is similar to the distribution in B2013a Figure 3c is our Figure 4a which presents our control case. This is because it also contains these unrealistic blocks in the valley.

4) P2 l1. The authors appear to be unaware that the bed topo in BedMachine v3 and that which was used in B2013a are essentially the same. The only difference is in the use of mass continuity near the margins where IPR coverage is poor. Any conclusions drawn in B2013a will be identical for BedMachine.

Our tests are focused on the effects of the removal of blocks in the valley in BedMachine v3. We are not presenting any results or discussion concerning differences between BedMachine v3 and the data used in B2013a.

5) P4, l35. Fig 5 does not show slope, it shows surface elevation. The slope in the interior of ice sheets is small everywhere. This is not the same as "near flat" which is meaningless. Eyeballing the Fig it looks like the canyon follows the surface slope quite closely in the interior. This sentence is a good example of the hand-waving vagueness that pervades other parts of the paper.

Surface elevation plots indicate slope through the distribution and concentration of contours. We have changed:

"Figure 5 indicates that the along-valley component of the ice surface gently slopes down-valley all the way to Petermann Fjord."

**6) P5, l5-6. I don't understand the logic of this statement. The canyon is not linear and doesn't follow the streamlines when it is not present (Fig 4a). This is nonsensical**

In this section we are trying to establish whether the current form and path of the valley is consistent with what could be expected for a subglacial river under the current ice sheet configuration. While the valley appears to intersect the east and west hydrological basins in the interior it does not follow the streamlines, particularly as it crosses NEEM zone. A perfectly aligned valley with present day streamlines is probably not to be expected given the different extents of the ice sheet in the past. B2013a Figure 3b is a good indicator of this as the basal hydrological divide is shifted to the west under LGM conditions and the valley still follows a path not far off the basal

hydrological divide. We have reworded and added some sentences discussing this issue and reference B2013a Figure 3b:

"However, while the valley appears to intersect the east and west hydrological basins in the interior it does not follow the water flux streamlines exactly, particularly as it crosses NEEM zone. South of the Petermann surface catchment the valley tracks roughly parralel, and to the west of, the basal hydrological divide, while Tributary projects towards the east of it (Figure 8). If the bedrock were flat, there would be only one basal water route towards the north and it would be directed exactly along the hydrological divide. Perhaps a subglacial valley perfectly aligned with present day basal water flux streamlines is not to be expected given the long period required to erode it and the different shape of the ice sheet in the past. As an example Bamber et al. (2013, Figure 3b) indicates that the interior basal hydrological divide is shifted to the west under conditions at the last glacial maximum. Nonetheless the valley still follows a path not far off the basal hydrological divide so it is possible that conditions favourable for subglacial down-valley water routing may have existed for a long time as has already been implied by the results of Bamber et al. (2013)."

7) Section 4 Discussion. I found this section far too speculative, hand-waving and non-scientific. The first part is a qualitative overview of subglacial water flow theory. The arguments for why water may be present in the canyon are OK, in general although the discussion of enhanced GHF was a bit muddled and unclear. It could have all been stated in half the space as basal frictional heating, warmer ice at depth are all well established concepts.

The overview on subglacial water theory is necessary to address the key issue of the paper on whether water flow within the valley is likely to be in the form of a subglacial river. Numerous changes have been made to the discussion and new results on the role of interior GHF are presented.

By page 7, 19 the discussion becomes too speculative.

Given the possible large extension of the valley catchment to Basin we think it is important to discuss this in this section. We try to make it clear in the discussion where we are uncertain.

The authors appear to be unaware that B2013a examined hydraulic potentials for the isostatically compensated bed (Fig 3A) and discuss this in some detail. In addition, the authors do not explain how 300-800 m of bedrock erosion from subglacial water flow is possible during the Holocene. That would be a challenge even for rapid basal sliding over a soft bed, neither of which is the case here. At LGM, ice flow and hydraulic routing was different due to larger ice sheet cover.

The LGM hydraulic routing issue is addressed above. The following has been added to the discussion on this erosion issue:

"The possibility remains that the present day valley form developed as a consequence of erosion under some or all of the following 1) under current conditions, 2) under last glacial maximum conditions, 3) during ice sheet retreat, 4) under reduced ice sheet cover, and 5) under ice free conditions. We simply do not have enough information. Tunnel valleys provide one demonstration of how subglacial water erosion can erode into hundreds of metres of bedrock. Given that the source is close to a proposed geothermal warm spot, past episodic subglacial down-valley discharges of water and sediment are a possibility. Much smaller amounts of erosion and deposition would be needed to maintain a base slope favourable for water routing, as is typical of rivers in general."

P7, l27. Why?

The concept here is that it seems unlikely that, in the absence of subglacial river erosion, a river formed when there was no ice sheet would take a form that would end up also being favourable for subglacial water flow under thick ice. In this sense the substantial changes to the form of the bedrock caused by the build up of km thick ice should make a roughly level paleo river valley capable of routing subglacial water all the way out from the interior, unlikely. Having said that the valley base does not appear to be perfectly level by any means, and does not follow the hydrological divide perfectly. We have added various points clarifying this issue to the discussion and Figure 8 to aid in explaining the points made.

9) P7, l30-32. This appears to demonstrate a complete lack of understanding of how the lithosphere responds to changes in ice loading. Why would there be a differential viscous response of the mantle across a < 10 km wide channel? In addition, how flat is the canyon and how flat is it after isostatic compensation? This is really incoherent.

We are referring to the along-valley differential viscous response, not the across valley one. A present day relatively level valley base over such a long distance is consistent with subglacial erosive and depositional water activity but does not prove it. The isostatic correction in B2013a Fig. S5 implies that on an ice-free Greenland the valley would be 600 m higher than present in the interior progressing to 200 metres higher near the coast. Today the valley base appears to be consistently between -300 and -500 metres with no clear trend along-valley. The following has been added:

"As stated earlier, the base today is not perfectly level as it appears to vary between -250 to -500 m but there is no obvious along-valley trend over its 1000 km length."

10) Section 5. As for the abstract, title and Discussion, this section needs to be rewritten, focusing on the actual results and not wild and unsupported speculation.

We hope that our responses address some of your concerns. We have made changes to the title, abstract, and discussion.

We refer to a the possibility of a long subglacial river in the title having considered the following points:

- The article is about the "possibility of a long subglacial river" and as such presents evidence for such a system while recognizing that the huge void in data precludes a conclusive result.
- The word "river" is appropriate when defined as "subglacial river" which has different properties to a river over open land. In addition, many "rivers" on Earth do not flow at all times, or flow at certain times only along certain sections.

Our definition of a subglacial river is clarified in the introduciton:

"The term "subglacial river" is used here to cover a variety of possible non-film subglacial hydrological conduit forms such as within R-channels, canals, Nye channels, or braided rivers. These different forms are explained further in the discussion. In addition, a "subglacial river" may incorporate storage within reservoirs along route that release water only over certain periods and can flow uphill in certain situations. A subglacial river beneath an ice sheet is therefore considered to have quite different properties from those of a terrestrial river."

**Response to Reviewer 2: Andy Aschwanden**

Thank you very much for taking the time to review our paper. We hope the responses below address the issues that you raised. *(Reviewer statements are in italics)*.

"First (as already pointed out by Jonathan Bamber), the title is misleading because Bamber et al (2013) [B2013a] already demonstrated the existence of a long subglacial channel. I suggest to change the title to something that more closely reflects the core of this manuscript. E.g. along the lines of "On the influence of a long subglacial channel on basal hydrology and ice flow"."

The title has been chosen because the focus of our investigation has been on the possibility of a currently active long subglacial river system along the valley route. We would prefer to keep the original title after considering the following points:

- The article is about the "possibility of a 1000 km long subglacial river" and as such presents evidence for such a system while recognizing that the huge void in data precludes a conclusive result.
- The word "river" is appropriate when defined as "subglacial river" which has different properties to a river over open land. In addition, many "rivers" on Earth do not flow at all times, or flow at certain times only along certain sections." We are going to add this definition to the introduction to be clear and we are considering adding an additional appendix that discusses potential discharge estimates. A definition has been added.

Regarding the originality of the study, B2013a did not demonstrate that an uninterrupted water pathway was possible under present day conditions from Interior to Petermann in the evidence they presented. Their figures show water pathways along numerous independent sections of the valley. They infer that the valley is probably continuous but do not present scientific evidence of the potential effects the continuity of the feature could have on the basal hydrology or ice sheet sliding. Therefore our results are original. We have added references to B2013a beyond what was already included to make it clear what their results showed, how they differ from ours, and how our results relate to inferences made in B2013a.

"The hydrological and ice flow modeling, which is the most novel aspect of this manuscript, is illconceived. The authors compare a simulation "Valley" to a "Control" simulation which to me has limited value. Why not compare the simulation to the readily-available surface velocity measurements, as one would hope that better physics leads to a closer agreement with observations?"

Thanks for the suggestion to compare with observations. Our goal in relation to ice sheet sliding is to provide an assessment of what impact an open valley introduced into a simulation can have. To do this it makes sense to compare two cases that are identical aside from the topographic modification, as we have done with "Valley" and "Control". The results therefore only show the consequences of the topographic change and so reveal the influence it has. For assessing whether the introduction of a valley improves the simulation with respect to observations it would be valuable to make the comparison you suggest, however this has not been our aim up to this point. All simulations have been rerun with a more advanced setup. The new results have a less marked impact on sliding and we have added the following statement to the results regarding this:

"Considering other uncertainties and inaccuracies, the differences are too small to allow assessing which case is better compared to observed surface velocities."

"In Figure 6 the differences between "Valley" and "Control" exceed 10 m yr-1. Looking at observation of surface speeds, I find no clear signature of a channel, contrary to what is expected. I wonder if this means that the sliding model is too sensitive to changes in basal water."

The simulations have been re-run with a shallow-ice--shelfy-stream scheme that captures the dynamics of fast-flowing grounded ice more realistically. The effects on sliding are less, however the increase in surface speeds over the valley are still present. It is possible that there is no subglacial river, or significant subglacial water within this section of the valley, i.e. it is mostly frozen to the bed. Alternatively since the model relies on a thin film basal hydrology it fails to model any type of subglacial river. Consequently it fails to channelize the water, potentially causing an unrealistic distribution of water in a wide film that has a greater influence on sliding. Generally the addition already mentioned is relevant here:

"Considering other uncertainties and inaccuracies, the differences are too small to allow assessing which case is better compared to observed surface velocities."

**"Furthermore, with a channel width of 5-10km, it is not clear to me how well a model resolution of 5km is able to capture the dynamics that the authors are interested in."**

This is a significant model issue and it is one that important to highlight. Unfortunately we are currently unable to simulate the Greenland Ice Sheet at higher resolutions than 5 km and this restricts our ability to correctly reproduce basal topographic features. We can only present the results we have and detail potential error sources. Judging by the basal water fluxes we present, we have successfully removed the blocks along the valley route, which was our goal. From a modeling perspective, this is a first step to demonstrate the potential extent of the impacts.

"Right now the manuscript consists of a lot of material that was already discussed by B2013a, an interesting but not well thought out modeling part, and a very speculative discussion on geothermal flux. While the manuscript has potential, I cannot recommend it for publication in its current form; a conclusion I did not reach lightheartedly. Maybe these suggestions can help to rewrite the manuscript, with focus on 1. what is novel compared B2013a 2. the ice flow modeling and how the inclusion of the channel improves agreement with the observed flow structure."

We present new scientific evidence that was not presented by B2013a and will reference B2013a further where appropriate. We hope that the points above help to address the issues presented on the modeling aspects. The geothermal heat flux discussion in relation to the NEGIS hot spot and the valley form are based on the published research of Fahnestock et al. (2001) and van der Veen et al. (2007) respectively.

**"Detailed comments:**

P2, L 1: Please note that in the interior of the ice sheet, BedMachine is based on kriging, mass conservation was only used near the coast. It is thus more or less equivalent to bed map of Bamber et al (2013) [2013b]."

This is noted as below. Our study is focused on adjustments made to open the valley rather than changes in the data since B2013b.

"However, based on the locations of the radar data lines that were used to generate this dataset and the limited extent of the valley bed elevation derived by mass conservation (two example regions are shown in Figure 1c,d), it is clear that these rises occur only in regions where data was not obtained."

"P2, L 11: change "when you consider", this is too colloquial".

We have changed this to "after accounting for".

"P2, L23: Maybe I don't understand the initialization procedure correctly, but why is 1990 the target date? P2 L 30 that corrections are made to bring the simulated ice thickness in agreement with observations. But to the best of my knowledge there is no 1990m DEM of Greenland available that can be used as a target?"

Our SICOPOLIS model setup is based on that used for the Ice Sheet Model Intercomparison Project for CMIP6 (ISMIP6). The reason why we chose 1990 as the initialization date for ISMIP6 is that the Greenland ice sheet was approximately in balance around that time, while its mass balance has become more and more negative since then. You are correct that there is no DEM for 1990. The nominal date for BedMachine Greenland v3, from which we also took the surface DEM, is 2007 <https://sites.uci.edu/morlighem/dataproducts/bedmachine-greenland/>.

However, the difference between the overall topography 1990 and 2007 is very small, so that it doesn't really matter. More important is that the climate forcing for the spin-up run ends in 1990 and produces an ice sheet that is very much in balance.

"P5, L5: "extremely gently" sounds awkward. Maybe just "gently" or "very gently"? P5, L 25: "where you get" is too colloquial."

We have changed "extremely gently" to "very gently" and have deleted "where you get".

"Figures: To increase readability, I recommend using the same color scales for bedrock elevation in all figures. Currently there a 3 color scales (Fig 1, 2, 5).

Thanks for this suggestion, This scales are now the same across these figures.

Fig 4: A close up of the NEEM zone would be helpful, the flow lines are hard to distinguish here."

Close ups of NEEM zone are now included in Figure 4.

**Marked up manuscript**

[revised manuscript text omitted]

---

## Author Response (AR2)

**Editors comments**

**Dear authors**

**The revised manuscript was reviewed by a referee. The referee judged that your rebuttals to retain the title and to introduce new terminology "braided river" are not valid, and suggested to change them. I echo this view and request a full consideration on these points. I think it beneficial to have speculative statements in the papers, even if there is a set of evidence that partly do not support speculations. However, such set of evidence in prior work should be adequately presented so that the readers can have the full knowledge base and develop a balanced view.**

**I would like to ask another revision to respond to the referee comments. Please provide a response letter as well as the marked manuscript, when you submit the revised manuscript. I will review the next manuscript by myself.**

**Kenny Matsuoka**

Dear Kenny Matsuoka,

We have decided to provide 2 versions of the manuscript for consideration. Based on the reviewer's and your suggestion that the title should be changed, we have changed the title in version 1 (200615_final_V1) of the manuscript as well as through most of the manuscript except when we are referring to consideration of prior research that has used this definition (see Appendix 1). The second version (200615_final_V2) retains the term "subglacial river" in the title and throughout the manuscript, and can be ignored if you consider it best for publication in The Cryosphere to change the title as suggested by the reviewer. The issue over the term "subglacial river" is an interesting one that we provide some detail on in the response to the reviewer, and in an appendix below. The reviewer has stated that the term "subglacial river" is not acceptable glaciological terminology or notation. Based on the previous research we have found that this term has been accepted as glaciological terminology by several journals for the same purpose of encompassing the variety of forms of channelized water flow within subglacial valleys under ice sheets. As such we believe the issue comes down to whether this terminology is acceptable for use in The Cryosphere. If it is considered acceptable then we would prefer for version 2 of the manuscript to be used, because this terminology is more likely to garner interest in this fascinating subglacial feature. To put it another way, children will not be inspired by the term "subglacial water channel" but may be by a mysterious "subglacial river".

Additional files are

**200120-200615_V1**: shows the changes between version 1 and the previously submitted version.

**200615_V1-V2**: Show the differences between version 1 (water channel) and version 2 (subglacial river).

Best regards,

The authors

**Response to Reviewer comments**

Reviewer comments are in ***bold italics***

***Re-review of Chambers et al***

***The authors have made an effort to address the referees' original set of comments and largely the changes have achieved this. The inclusion of Fig 7, in particular, provides greater context for the more speculative elements of this paper regarding the presence of a continuous active hydrological channel along the valley identified in earlier studies. There is, however, still a considerable amount of speculative material and sweeping statements that would benefit from improvement.***

***Specific details***

***Title. The authors have argued to keep the original title (with some small changes) rather than focusing on the impact of the modified bed topography on water routing and ice motion. They have also decided to retain the use of the term "subglacial river". The former is OK but the latter I find problematic. There are many papers on subglacial hydrology and several nice reviews. One of the most recent focusing on Greenland is [Davison et al., 2019]. This and all the other articles I am aware of use accepted glaciological notation and definitions. I can find no reference to "subglacial rivers" and, given the extremely speculative nature of the paper, it seems entirely unjustified to introduce a new term or concept with no theoretical or empirical foundation.***

In version 1 of the manuscript we have changed the title to address the key point while version 2 retains the title, however it is our preference to keep the original title based on the points outlined below, and if it is deemed appropriate terminology for The Cryosphere.

The reviewer outlines their reasoning for their request to change the title. Because of this we can assess whether the reviewer makes a valid scientific argument for changing the title.

The reviewer presents an objection to the use of the term "subglacial river" because:

1) the reviewer did not find any prior reference to "subglacial rivers",

2) to the reviewer's knowledge, it is not an accepted glaciological notation or definition,

3) and as the authors, we are introducing a new term.

We have looked into these arguments to determine their validity. The term "subglacial river" has been used in the scientific literature many times before and has been specifically used in reference to under ice sheet water channels. Some relevant references that specifically refer to "subglacial

rivers" or "subglacial river valleys" (e.g. Mooers, 1989; Clayton et al., 1999; Remy and Legresy, 2004; Popov and Masolov, 2007, Klokočník et al., 2018) have been added to the manuscript. Subglacial streams and rivers are well known under glaciers, with subglacial streams defined as "A meltwater stream that flows underneath a glacier or ice cap." by the Oxford reference (https://www.oxfordreference.com/view/10.1093/oi/authority.20110803100539533). A stream is defined as "a body of running water (such as a river or creek) flowing on the earth" by Merriam Webster (https://www.merriam-webster.com/dictionary/stream). A river is simply a large stream and that is why the previous papers have referred to "subglacial rivers" rather than "subglacial streams". For a 1000 km long watercourse we consider the term "subglacial river" more appropriate than "subglacial stream" and based on the previous research, the term has already been publised as accepted glaciological terminology. Since this is raised as a fundamental issue with our manuscript, and its title, we are including some quotes from prior references to subglacial rivers and subglacial river valleys. The quotes are below in Appendix 1.

***The authors also refer to "braided rivers" (l27 p 2). This is also a term that is new to the field of subglacial hydrology.***

We have removed the term "braided river" from both versions of the manuscript.

However the reviewer suggests that the term "subglacial braided river" is new to the field of subglacial hydrology. There are, however, a number of examples in the literature of references to subglacial river braiding. Nonetheless to avoid further complication we have removed the term "braided river" from the manuscript.

***The authors provide no basis for introducing these new concepts in subglacial hydrology.***

Since the concepts are not new we have added some of the references to the prior work in the introduction.

***The justification given on p2 is entirely inadequate and inappropriate. The section in the discussion, already commented on in the previous reviews, simply rehearses in a qualitative way established concepts in subglacial hydrology. The term "braided river" is not discussed anywhere. If the authors wish to introduce new terminology into the discipline they need to provide a justification. None exists and they should use accepted terminology to avoid confusion, ambiguity and misleading readers.***

Both terms have been used in previously published research, though we have removed the term "braided river" completely from both versions. In the section referenced we have added the relevant references.

***They should replace terms such as "subglacial river channel" with "subglacial water channel" and remove reference to a river from the m/s. The title would then replace "subglacial river" with "active subglacial water channel"***

The term "subglacial river channel" has been removed from both versions of the manuscript. Given the number of prior references to "subglacial rivers" under ice sheets, we have not completely removed the term "subglacial river" from either version of the manuscript but rather have added the references. However in version 1 the term is mostly replaced by the term "subglacial water channel", including in the title.

**P2, l11, l20 and elswhere. Data is plural.**

These changes have now been made.

**P2, l16. The authors claim there is a continuity of the signature of the valley on the ice sheet surface where the bed topo has a valley. This requires at the minimum a citation to a study that demonstrates this although I do not believe such a study exists. In fact, Fig 2 of [Ekholm et al., 1998] suggest the exact opposite. The surface expression of the "valley" is intermittent and aligns well with the bed topo valley locations. In other words, the surface expression suggests the contrary: that the "valley" may not be continuous, or at least, it's depth and width may not be sufficient to produce a unique surface expression everywhere.**

We have removed this sentence.

The continuity on the surface of the ice refers primarily to the section through the coastal mountains where the signature is very distinct and continuous whereas the Bedmachine data includes large blocks in the valley due to data interpolation. In order to demonstrate this properly an additional figure would be required. Over the thicker ice in the interior, the signature is harder to determine, affected as it is by the thickness of ice, the variable form of the valley, and the direction of ice movement.

 **Further, fig 4 from the same paper indicates that the depth of the valley, where observed, is highly variable over short distances: 100 m deep and narrow in fig c and 200 m deep and twice the width in Fig b around 30 km upstream.**

The reviewer appears to be referring to the degree of incision/form of the valley in the bedrock which is not the same as the depth in terms of the absolute elevation (in relation to sea level) and is not relevant to the issue of whether the base of the valley is roughly level. We have earlier referred to the fact that the valley takes a variety of forms along its length. In fig 4b and c referred to by the reviewer (from Ekholm et al. (1998) and included below) the valley base elevation is almost the same with both close to -300 m. Therefore based on these two examples, the base in this section in fact exhibits almost no variability. It suggests that despite the high variability in the from of the incision of the valley into the bedrock, the elevation of the base varies very little.

[Figure]

**Figure 4.** Bedrock topography, relative to WGS84, (black) and high-pass filtered surface elevation (red) of four IPR-profiles in north Greenland anomaly region from south (A) to north (D). Smaller plots of surface groundtracks are inserted for identification purposes (see Fig. 3).

*Assuming a fixed depth of 400 m will, by default, result in water being routed along the valley because it is approximately aligned with the surface slope.*

*The authors do investigate the sensitivity to assumed depth in Fig A3 and the water routing, not surprisingly, is sensitive to this. The IPR data indicates, however, a variable width and depth valley. This is also confirmed in Fig 2 of [Bamber et al., 2013] that suggests a valley that ranges in depth from ~500 m to <100 m.*

The reviewer is again referring to the degree of incision into the bedrock rather than the elevation of the base of the valley. The reviewer is correct that there is a high degree of variability in the form of

the valley which is the reason why we have referred to it as a valley rather than a canyon. However the cross-sectional form of the valley is not relevant to our discussion, rather it is the elevation of the valley base that we argue is relevant. In Bamber et al. 2013 Fig 2 the valley base is -280 m, -380 m, and -450 m so it varies by 170 m, across these three cross sections. This variability is discussed in the discussion on page 10:

"As stated earlier, the base today is not perfectly level as it appears to vary between −250 to −500 m but there is no obvious along-valley trend over its 1000 km length."

Our experiments act to level the base of the valley, so the degree of incision varies depending on the elevation of the surrounding bedrock. This is also a determining factor of the degree of incision in reality.

*It would be useful for the authors to acknowledge the idealised nature of their "thought" experiment and that it is not designed to replicate the observed characteristics of the valley and is just that "a thought expt", which is fine as long as the m/s reflects this appropriately.*

Have added "as a thought experiment" to the abstract and the conclusion and to the goals section.

The manuscript includes a number of other sections referring to the uncertainties in the study.

*P11, l29-30 "However, since water has already been detected in the valley (Ekholm, 1998). " This is too firm a statement and needs to be rephrased to something more accurate such as "since water has been inferred to be present from IPR data in a limited number of locations"*

This has been changed.

*Refs*

*Bamber, J. L., M. J. Siegert, J. A. Griggs, S. J. Marshall, and G. Spada (2013), Paleofluvial Mega-Canyon Beneath the Central Greenland Ice Sheet, Science, 341(6149), 997-999.*

*Davison, B. J., A. J. Sole, S. J. Livingstone, T. R. Cowton, and P. W. Nienow (2019), The Influence of Hydrology on the Dynamics of Land-Terminating Sectors of the Greenland Ice Sheet, Frontiers in Earth Science, 7(10), doi:10.3389/feart.2019.00010.*

*Ekholm, S., K. Keller, J. L. Bamber, and S. P. Gogineni (1998), Unusual surface morphology from digital elevation models of the Greenland ice sheet, Geophys. Res. Lett., 25(19), 3623-3626.*

**Appendix 1– list of example quotes from prior papers referencing "subglacial rivers"**

The reviewer has stated that the term "subglacial river" is not acceptable glaciological terminology or notation. Below are example quotes from previous published research that refer to "subglacial rivers" or "subglacial river valleys". These example have been found through a brief and limited investigation and so likely does not include other important references to "subglacial rivers" or to the first or most recent published article that refer to "subglacial rivers".

**1.) Mooers 1989: Quaternary Research**

"Numerous subglacial river channels associated with Late Wisconsin glaciation throughtout the upper Midwest"

page 9:

"Moreover, the vd values (together with Fig. 2b–d) indicate a possible connection between LV and the lake 90E area throughout some ==subglacial rivers==."

"Literature (Sect. 1) speaks about a possible mutual "underground" connection and interaction among the lakes (by ==subglacial rivers==), here of LV, 90 E, S and the others nearby (Sect. 3.4)."

"The subglacial rivers in this area, a "gravitational footprint" of those we demonstrate in Fig. 3 c, d, and g, would be realisation of a connection between the lakes, may be working from time to time."

From page 12:

"If we follow this course of the ==rivers== to the west, we arrive into a basin (Fig. 3 a, d, g) almost as large as the LV area where several smaller lakes were indicated (Sect. 3.5). We propose to call this basin as "90 E Basin" (because its largest lake would be the 90E lake) and the ==hypothetical river "Middle River"==. Another less developed W-E river structure is indicated by blue anomalies (Figs. 2b and 3b) at the western edge of Lake Vostok. The lake could be fed even by a broad lowland belt west of the lake, but the Middle River structure would represent the longest geophysically observed anomaly of the valley type."

page 15:

"If one follows the course of the ==subglacial river==(s) to the west of LV one arrives at a basin (Fig. 3 a, d, g) almost as large as the LV basin where several smaller lakes are indicated; we call this basin

"90 E Basin" (because its largest lake would be the 90E lake) and we call the main hypothetical river "Middle River""

[revised manuscript text omitted]

---

## Author Response (AR3)

Dear Kenny,

Thank you for the time you have spent on a thorough review of our manuscript. We have attempted to address the points and concerns and have revised the manuscript quite a lot. Please find the specific responses in *italics* to your comments below.

Best wishes
The Authors
* * *
Editor Decision: Publish subject to minor revisions (review by editor) (22 Jul 2020) by Kenichi Matsuoka
Comments to the Author:
Dear authors,

Thanks for submitting the response letter together with two versions of the manuscript for further consideration. The main difference between these two versions is whether the feature is referred as "active subglacial channel" or "subglacial river". The authors made a point that "subglacial river" is an accepted glaciological term in the response letter so I evaluated the latter version for the further consideration.
I made a careful review for the final judgement and found a large number of points that need to be considered before the acceptance of this work in its final form. Please consider following suggestions and address questions in the next revision, and provide a marked manuscript and response letter as well. Page/line numbers refer those in the marked "200615_final_V2" manuscript.

General comments:
- Title: as I said above, the authors convinced me to keep using subglacial river. However, I don't think that the current title captures the main point of this study. The possibility of subglacial river largely depends on a continuous topographic valley, which is hypothesized in this study. My suggestion is "Possible impacts of a 1000-km-long hypothetical subglacial valley towards Petermann Glacier in northern Greenland".

- *Thanks for the suggested title change, it has been changed with one addition being "subglacial river valley" is used instead of "subglacial valley". The motivation of this study was to investigate the possibility of a long subglacial river and if we remove "subglacial river" from the title then it does not represent the manuscript's focus as well. However if this is not acceptable "river" can be removed without further objection.*

I am open for other alternatives, but please be careful to clarify that this is a sort of thought experiments and assessments of a hypothetical subglacial valley which does not appear in the recent bed topography dataset but cannot be denied because of the inadequate data coverage. Also, geographical scope/interest of this study is somewhat unclear at the beginning but gets clearer in Results. Please clarify the geographical scope of this study somewhere in Introduction so that it is clear for the readers from the beginning.

- *P2L3: Added to first paragraph: "The valley extends from 74.18 ◦ N, 42.54 ◦ W in the Greenland interior to 80.21 ◦ N, 56.46 ◦ W under Petermann Glacier. As described in (Bamber et al., 2013), the valley may extend further southward but a sparcity of radar passes makes this harder to determine.*

Also I found both Petermann Glacier and Petermann Ice Stream in this manuscript. I am not a Greenland specialist, but I think Petermann Glacier is more common than Petermann Ice Stream. In any case, use only one of them consistently throughout the manuscript.

- *Removed all references to "Petermann Ice Stream". Replaced with upper Petermann Glacier where needed.*
- *Have added "Upper Petermann Glacier" labels to Figs 1 and 6 to clarify.*
- *Referring to upper Petermann Glacier as the section of the glacier above where the valley enters to Petermann Glacier.*

- Another general suggestion is to re-organize Introduction, Section 2.3 and Appendix A so that the motivation of this work and bed modifications can be better presented as a single information package. The current manuscript presents this point at three different locations, and each of them is neither comprehensive nor standalone. A possible solution is to make Introduction shorter, and change Section 2.3 to a new Section 2 "Limited knowledge of bed topography" or "Bed topography: current knowledge and hypothesis" (I'm open for other section titles) before new Section 3 "Model and methods". In this way, the bed topography is explained before the current Section 2.2 "basal sliding and hydrology".

- *P2L17: Created new section 2 "Limited knowledge of bed topography"*
- *Combined introduction bedrock section, Appendix A and section 2.3 into section 2*
- *Figures have been re-arranged to account for these changes.*

- I cannot accept the majority of Discussion for the publication, because they are relevant to the main results only to a small degree, or they are speculative based on thought experiments. Please revise Discussion largely; shortening this section does not reduce the value of this paper. Rather it helps readers to get the main points of your paper and stay focused.

- *The discussion has been changed following the comments below.*

Line-by-line suggestions and questions:
P1L1: Re-organize the beginning of the abstract so that it does not start with a question. For general readers, it is hard to immediately understand thoughts behind this question. It is better to explain the current status of our understanding of bed topography (including its limit), and potential impact of the hypothetical valley on ice flow and mass balance of the ice sheet. After stating these points, this question can be better understood.

- *P1L1: The question is removed and the bed topography issue is introduced first.*
- *Removed references to "sediment flow" from introduction and throughout manuscript.*

P1L2: This is only one place you say Petermann Glacier, and you say Petermann Ice Stream at other locations. Please sort it out.

- *"Petermann Ice Stream" has been removed from the manuscript.*

P1L13: Is "present day active long subglacial water river" observed? I cannot catch what you mean exactly.

- *It is not observed but rather we are attempting to determine if it is possible that a currently active subglacial river is possible along the valley.*
- *"active" has been removed from manuscript.*

P1L20: "mirror like" radar reflectors are expected only when there is a subglacial lake. Subglacial water channels with a smaller spatial extent do not necessarily make "mirror like" reflectors.

- *The finding of Ekholm et al. we belive is written correctly here and the reference is to "not mirror like" returns that they found.*

P2L6-: Mention the fractional length of the mass-conservation-derived BedMachine topography to the entire hypothetical valley. E.g. "The mass conservation method was applied to the lowest one sixth of main flowline towards the Petermann Glacier, so the majority of the bed topography in this region is derived by Kriging so that its error largely depends on the location of radar data."

- *P2L19: Thank you, this sentence has been added.*

P2L31: change to "a possible continuous subglacial valley"¨

- *P2L13: Change made.*

P3L3: at the beginning of this paragraph, add the spatial resolution of the model (5 km). You said "as detailed above" at P5L7 but you didn't show it actually.

- *P3L7: The spatial resolution is stated in the second paragraph. We have now added it also to the beginning of the first paragraph as requested.*

P3L4: geographical model domain is uncertain. Is it better to say "….the state of the entire Greenland Ice Sheet"? (or is the simulation made only for the northern Greenland? Anyway, please clarify).

- *P4L5: Added "entire".*

P3L7: change "final" to "last"

- *Change made.*

P3L10: Do not cite a manuscript in preparation in a way like (Greve et al. 2020). As a rule of thumb, manuscripts in preparation cannot be cited.

- *Sorry about this mistake. This manuscript is now published online but was not at the time of this review. The reference is corrected*
- *P4L10: Goezler et al. 2020 has now been accepted for publication and we have changed it to "in press"*

P3L11: change to "computed ice topography"

- *P4L11: Change made.*

P3L34: local depressions are not necessarily related to subglacial lakes. If the authors refer only to subglacial lakes, then this filling can be made only if the flat part is larger than a certain extent (i.e. lake extent). I assume that the authors rather refer to any type of small, local depressions in the bed topography.

- *P4L30: Removed "to account for subglacial lakes"*

P4L3: spell out RMSD and I think modeled and observed surface flow speeds are compared both in the logarithmic scale (rather than only observed speed in the logarithmic scale).

- *P5L1: Changed to root-mean-square deviation. Added comma to clarify sentence.*

P4L4: Remove a citation to a manuscript in preparation.

- *Updated as above.*

P4L9: typo? "Petermann in at an average…"

- *Corrected.*

P4L10: It is said that the modification is made to make the average elevation around -400 m. However, it is unclear for me why the presented procedure gives this average depth. Probably, the authors aimed to say that the derived hypothetical bed topography gives the valley floor at about -400 m. However, as far as I can see in Fig. 2a, BedMachine gives slightly shallower valley floor (it is represented as yellow/brown, instead of distinct green for your hypothesized valley). So, I doubt that this procedure gives the bed elevation of -400 m. Did you make any other adjustments and if so why is this floor depth of -400 m chosen? Overall, I would like to see Section 2.3 improved significantly to present the bed topography modifications more clearly. Some suggestions follow.

- *We have removed the reference to -400 m, it was a rough guide but through the process used it did end up varying quite a bit along the route as you state.*
- *P3L9 This paragraph has been updated following the suggestions below*

P4L10: Does "standard topography" refer BedMachine topography? If so, please say so explicitly rather than make a new term.

- *P3L6: Changed to "unaltered BedMachine topography down-scaled to the 5 km grid ".*

P4L12: Change to "consequences of the modified bed topography".

- *P3L8: Changed*

P4L14-16: Show the size of the valley polygon (how wide is the polygon from the valley center line? Might be 5 km, the resolution of the model??). Later at line 17, this polygon is converted to the center line. In principle, polygon (with a certain areal extent) cannot be converted to a flowline. Development of the modified bed topography is subjective by nature, which I can accept, but please be careful to explain the process as clear as possible.

- *P3L11: Changed "a polygon of the rough location" > "a polyline of the rough location"*
- *The "polygon" changed to "drawn polyline localizing" the valley is converted to a "smooth" centerline ", and then described through a spline" changed to "though spline interpolation". Data points situated within the valley "polygon" changed to "buffer" are converted from their Cartesian coordinates (x;y) towards this flow-oriented coordinate (s;n) system.*

P4L16: What is "a new BedMachine"? Do you refer a modified bed topography, i.e. "Valley"? Please clarify.

- *P3L12: Changed to "used to interpolate a new modified bed topography to be used in the "Valley" simulation"*

P4L16: If I understand correctly, the authors also used kriging (i.e. same procedure and parameters of BedMachine), which the authors criticized in Introduction. Is it correct? Then except for the saddle removing step, what do the procedures presented in the second paragraph bring in this study? Why do you need the steps, instead of simply removing rises from BedMachine?

- *Outside of the domain of the buffer we used the same Kriging parameters as BedMachine. Inside the buffer we used anistropic kriging, as knowledge of the thalweg is included through a morphed coordinate system.*
- *P3L19: Sentence added:*
- *"Moreover, outside of the domain of the buffer we use the same Kriging parameters as BedMachine, whereas inside the buffer we use anistropic kriging, as knowledge of the thalweg is included through a morphed coordinate system."*

P4L28: Re-consider the title of Section 2.4. I cannot see why it is "idealized". This section can be probably named as "Sensitivity tests on geothermal flux" or "Possible melt water supply from NEGIS".

- *Removed idealized.*
- *P5L5: Changed to "Sensitivity tests on geothermal flux"*

P4L29: probably "whether" instead of "where" (you compare only two experiments), and add "… is routed to this valley." Do both Control and Valley bed topography route the subglacial water to this valley? Or do you enforce such water flow in the model?

- *P5L6: Changed to "whether basal water melted near the source of NEGIS is routed down the valley."*
- *We do not enforce the water flow, we just insert the spot of enhanced geothermal heat flux.*

P4L33: Fig. 7b cited here, though only Figs. 1 and 2 are cited to this point. All figures need to be named in the order of citation in the paper.

- *Figures have now been reordered.*

P5L1: Does Fahnestock et al. infer 1.5 W/m2 at the hot spot? If so, please say so and if not please explain why it is set to 1.5W/m2.

- *Ours is a bit higher peak heat flux, we explain:*
- *P5L10: To do this a bell-shaped geothermal heat flux anomaly centred at 74 ◦ N, 40 ◦ W is introduced with a 1-sigma radius of 50 km and peak heat flux of 1.5 W m −2 which creates a broader region with a comparable value to the regional heat flux of 0.97 W m −2 estimated byFahnestock et al. (2001).*

P5L5: this section shows (1) Control, (2) ControlIS, (3) Valley, (4) ValleyIS sensitivity tests. However, at P5L32, sensitivity tests are mentioned again and cites Appendix B. Appendix B presents four more sensitivity tests, not 1-4 above, which is very confusing. Please re-organize this section and mention additional four sensitivity tests in this section. And more fundamentally, do you really need additional sensitivity tests presented in Appendix B?

- *The sensitivity to valley depth tests were requested during peer review and required extra work so we would rather not remove them. We believe they present useful results on how sensitive the routing is to what we set the depth of the valley to.*

- *The wording is changed to more clearly distinquish these tests from the geothermal heat flux tests.*
- *P6L13: "Four additional sensitivity tests have been used to investigate the robustness of these responses by altering the valley depth (Appendix A). These additional results indicate that the location and magnitude of this water flux out of the valley is sensitive to the valley depth due to its consequent effect on the steepness of the valley sides."*

P5L7: you didn't say the resolution of the model (5 km), though you say "as detailed above" here.

- Addressed earlier.

P5L9: I think it clearer if you say "basal water thickness"

- *Changed to "basal water thickness" throughout article.*

P5L10: Is it better to say "the simulated basal water depth is affected by the Valley bed topography not only in the Petermann catchment but also the North Greenland in general." I think most readers understand that this paper primarily focuses on the Petermann catchment, and it is suddenly said North Greenland here.

- Changed to:
- P5L21: "For the Petermann catchment and northern Greenland in general"

P5L11: better to say "separated" instead of "independent", and change "deeper basal water" to "thicker basal water". I already pointed out above, but also change "basal water depth" to "basal water thickness", and "deeper and uninterrupted" to "thicker and uninterrupted". I don't point out this "deeper vs. thicker" issues below but the same suggestion is applied for all locations.

- *Changes made and throughout article:*
- *Basal water "depth" changed to "thickness" and "Deeper" changed to "thicker"*
- *Changed label on Fig. 7.*

P5L14: I don't call a water body of 10 cm thick as subglacial lakes. It is very unclear to show the thickness as "larger than" (even 100 m is larger than > 0.01 m!). It is much clearer if you say something like "thicker water bodies of about 10-15 cm were found at 6 locations along the valley". (Obviously I made up 15 cm and 6 locations!).

- *"subglacial lakes: removed.*
- *Changed sentence to:*
- *P5L25: "The thickest water (> 0.01 m) along the valley route occurs at 7 locations where the Priority Flood algorithm has been activated to a maximum thickness of 10 m."*

P5L15: do you mean no difference between Control and Valley experiments by "no change"? The word "change" often refers temporal changes, so please describe it in a more explicit way.

- *Changed to:*
- *P5L26: "In most interior areas away from the valley there is little or no difference in the basal water thickness between Control and Valley."*

P5L20ff: It seems to me that the location of thicker water shifted eastward at 74-75N as the bed topography is modified. Is it directly associated with the shift of valley position? I think it is better

to mention this in the text. What features are directly related to the hypothetical topography and what features are more "wow, it's unexpected"?

- *The valley position is not shifted but rather opened along route. More water is pirated by the valley hence the reduction seen to the west at 74-75 N.*
- *P5L32: Have added "an indication of water piracy by the valley."*

P5L26-27: I cannot read "stuck" features from Fig. 4.

- *Have removed this part of the sentence, it is a little difficult to see unless the figure is made very large.*

P5L29ff: It is very hard to follow these statements. I think that the authors are trying to mention the feature of a "blue-red" pair blowing from the valley to northwest in Fig. 3d. Do I understand correctly? Then why don't you explain it when you discuss Fig. 3 a few lines above?

- *It is mentioned when we discuss now Fig. 6. as quoted below (includes additional upper Petermann Glacier definiton now):*
- *P6L2: "The one region of increased basal water outside of the valley is a region that extends towards upper Petermann Glacier from NEEM zone in Figure 6d. NEEM zone is defined as the area of the valley to the southeast of the NEEM ice core drilling site as indicated on Figure 1. The effect of these changes is to redistribute the basal water under upper Petermann Glacier into a narrower and thicker plume that can also be seen in Figure 6b.*

P6L2: Is 1.6 m3/s a simple conversion of 10,000 m2/a (check the unit)? Also why do you convert it? Are you trying to tie this water flux to some observations at NEEM?

- *Here we attempt an estimate of the discharge using units typically used for river discharge.*
- *No, unfortunately there are no observations for discharge within the valley to compare with near NEEM.*

P6L5ff: What is the point of Fig. 5? Fig. 4 shows water flow directions, which are calculated based on the subglacial hydraulic potential I believe. Then why do you want to show ice and bed topography (i.e. more indirect way to show the water flow direction) here? If you argue that Fig. 5 is necessary in addition to Fig. 4 and others, consider presenting it in a smaller size. Also, is the bed topography presented here Control or Valley? Clarify it in caption.

- *The purpose of now Fig. 8 is to show how the path of the valley is related to the ice surface topography, how it roughly follows the northward ridge, is continually downslope. This figure allows the reader to see the broader picture of how the valley is related to the ice sheet shape.*
- *We are very keen to keep this figure if possible. It is important that the figure stretches north to south from Petermann to Basin. The resolution can be reduced but otherwise do not really want to reduce the geographical scope of the figure.*
- *Have added "BedMachine" to the caption.*
- *Reduced size, now it's about half the size.*

P6L9-L14: I cannot follow the argument here. Why can't you conclude that the bed is sloping or flat simply from the bed topography? In what context is post glacial adjustment discussed here?

- *P6L25: Post glacial adjustment sentences have been removed from here.*

-

P6L15ff: I am puzzled. Fig. 2 shows no topographic modification in the Steensby Glacier area (forgive me if I call it wrongly, but I want to point a glacier east of Petermann). No topo modification and no increase of water thickness in Fig. 3c are consistent, but the significant increase of surface flow speed (Fig. 6) is inconsistent with the other two. Please explain. It seems that this point is discussed and concluded negligible (P6LL23-24), which is incomplete and not convincing. Also, as the bed topography is modified, ice thickness is modified. Then the difference in surface flow speed is a combined effect of increased (thicker) ice deformation and basal sliding. Is it more straightforward to examine ice velocity of the deepest modeled ice layer?

- *P6L27: Have altered this paragraph based on this comment. We do not at this point know the reason for the sliding increase around Steensby and Ryder Glacier.*
- *P6L31: "These increases are consistent with the redistribution of basal water seen in Figure 6 and could also be related to the ice thickness change due to the modification of the bedrock."*
- *"Sliding changes also occur at certain outlet glaciers along the west coast as well as on the north coast around Steensby and Ryder Glacier to the east of Petermann. These changes, away from the valley and Petermann, are inconsistent across different model setups and do not show an obvious relationship to the changes in basal water distribution. They may be related to knock-on effects from the redistributions of ice mass due to the inserted valley. Considering other uncertainties and inaccuracies, the differences are too small to allow assessing which case is better compared to observed surface velocities."*

P6L27ff: I am afraid that Figure 7 needs to be revised to present these points clearer (or the authors read Fig. 7 too much). The most obvious thing is that basal melt rate along the valley increases only if the Valley topography is used. And regardless of the topography, the additional NEGIS hotspot increases basal melt rates towards the NEGIS and towards the west. I cannot immediately see the differential amount of basal melt towards NEGIS and west, when Control and Valley topography results are compared. Maybe the authors found this west-east contrast by examining the modeled results, but Fig. 7 is in any case inadequate to make this point. Also, this and following paragraphs in this section include discussion in addition to results. Please distinguish discussion and results more clearly and move statements to discussion as appropriately.

- *There may be a misunderstanding here.*
- *The basal melt rate is shown in the left panels and the basal water thickness differences are in the right two panels. So in the left panels there is no increase in basal melt rate along the valley shown in this figure.*
- *The most important point to be made here is in the right panel. This shows the difference in basal water thickness between two simulations with the same topography, but with one having the increased basal melting only in the region of the hot spot. So it shows that water thickens all along the valley which indicates that water flow from the hot spot down the entire valley is occuring. This is a key result demonstrating this fact and the potential for water flow all the way down the open valley.*

P6L35: Please carefully check the statements in this paragraph, and its evidence presented in the figures. I cannot agree with this statement about the region north of 80N.

- *In Figure 7d in the region north of 80 N the only region that has a significant regional change in water thickness is in the Petermann region.*

P7L1-3: You discuss the water flux here, while you have discussed basal melt rate in this paragraph. Reorganize.

- *The paragraph discusses basal water thickness primarily.*

P7L4ff: The majority of the discussion here is irrelevant or at least not significantly connected to the results presented here. In particular the first few paragraphs explain general knowledge, which is not necessary for this paper in my opinion. I suggest to delete the first 1-3 paragraphs and the 5th and 6th paragraphs because of inadequate relevance to the results. The 4th paragraph is somewhat relevant, but it is not directly linked to specific results in the current manuscript. If you retain this paragraph in the new version, please deepen the discussion and clearly connect the discussion to the results you obtained.

- *Deleted the first 1-6 paragraphs of the discussion.*

P8L16ff: I am not yet convinced that this discussion and Fig. 8 are necessary in the final form of this paper. I see the importance to examine boundaries of surface/ice and subglacial/water drainage basins. The current Fig. 8 shows the subglacial/water boundary for the Valley topography (I assume so, but it is not clear in the caption). If the authors have important discussion points on this issue, please present both boundaries of surface/ice and subglacial/water drainage basins for (1) Control topography and (2) Valley topography in two panels next to each other, and discuss how (2) is different from (1). Keep it in your mind that topography modification is hypothetical and the main aim of your work is to draw an attention to improve the bed topography of northern Greenland. You don't need to examine all small features.

- *A colour bar has been added and the topography has been removed from the plot to make it clearer.*
- *The last reviewer said that figure 8 was helpful so we would prefer to retain it, hopefully now it is a bit more clear with the topography removed.*
- *The caption states "basal water flux from the sensitivity simulation that has a valley with fixed depth of −100 m (valley removal described in Appendix B)"*

P9L4ff: I think that the discussion about the link between the NEGIS hotspot and Petermann Ice Stream is valid. However, this paragraph starts with the discsussion about R-channels and so on. Please re-consider this paragraph and make sure that the discussion is directly pointed to the most important issues.

- *Removed "The current simulations do not include subglacial channelized water flow, such as within R-channels, in the hydrology module. Channelized water flow could funnel different amounts of water away from, and to, particular locations leading to focused areas of suppressed and enhanced sliding. This may effect the sliding model results that should be seen as a view of the potential for change in different regions rather than a prediction for the valley's influence on sliding speed."*

P9L17ff: I don't really think that discussion in this and following paragraphs are directly relevant to the hypothetical subglacial valley. Nonetheless, if you really want to point out these issues, limit the topics which are directly relevant and highly important for the ice-sheet dynamics. I really think that the 4-page-long discussion in the current form can be dramatically shortened to 1 page or so. It actually helps the authors to address the most important issues directly related to the results.

- *P8L21: This paragraph has been shortened to 2 sentences.*

P11L6ff: Please update this section according to your revisions in Discussion. The link to the NEGIS hotspot can be mentioned here, but the statements to this point (P11L16) can be shortened. I didn't point out at other locations, but the authors occasionally call the feature as "active" subglacial river, though the word "active" was removed from the title. I don't clearly see meaning added with this word, so suggest the authors to carefully consider what it really means. Also, the statements about wet sediment flow system and paleo-fluvial rivers are too weak to be concluded/presented here. My take-home message from this paper is that more radar surveys are necessary to examine the absence of these "removed" rises. If this is the case, please clarify this point both in Abstract and conclusions, and possibly in Introduction.

- *P9L25 we think has been addressed earlier regarding the significance of the result that water melted in the deep interior can pass the whole way along the valley if it is opened.*
- *remove "active" in relation to subglacial river throughout the manuscript.*
- *P10L5: The paleo-river is referring to terminology used by Bamber et al. (2013) so we are making the point that a subglacial river requires a different understanding and form to a river that flowed when there was little or not ice because they are such different systems.*
- *Referall to wet sediment flows have been removed.*

P11L9: change "are not real" to "may not be real". The lack of evidence does not means that there are no rises.

- *P9L3: Changed to may not be real.*

P12L3ff: Overall revisions are necessary to the two paragraphs for Appendix B. For example, it is unclear what "linear" and "idealized" mean (L5), why 26 valleys are used for 4 additional experiments (L5), what "grid point values" of "along-valley rectangle" mean (L6), what "short SICOPOLIS simulation" (L7) and "standard SICOPOLIS topography" (L9) mean, and what "4th test" refers (L9). There are just examples to show why I require overall revisions to Appendix B. I expect concise, but adequate descriptions in Appendix. Also, above all, do you really need Appendix B?

- *P10L9 onwards:*
- *We have changed the wording and added some additional sentences, specifically:*
- *Changed "linear" to "straight"*
- *Changed "valleys" to "channels"*
- *Changed "grid point values" to "bedrock elevation grid point values"*
- *"short SICOPOLIS simulation" is removed.*
- *"The first three of the four tests insert 26 straight 10 km wide channels to form an uninterrupted valley from Interior to Petermann. The channels are created using the Matlab function "inpolygon" that sets bedrock elevation grid point values within a rectangle that is aligned along the path valley. This method is used as it creates a flatter base than the method used in simulations Control and Valley. Consequently it is more suitable for testing the sensitivity to a particular valley base elevation."*
- *Changed "standard SICOPOLIS topography" to "unmodified BedMachine topography"*
- *We have added additional explanation about the "first three of the four tests" to clarify about the "fourth test"*
- *Appendix B is now Appendix A and was introduced to address earlier reviewer comments. It is useful for showing the sensitivity of the basal water distribution to the valley depth given that the form of the valley is uncertain so we would rather keep it.*

P.17 Figure 1: the legend says Petermann surface catchment, but I think it is a buffer of 200 km used to modify the bed topography, rather than defined from the surface topography. Also, the label shows Petermann Glacier but the text mostly refers as Petermann Ice Stream.

- *The Petermann surface catchment is correctly labeled.*
- *"Petermann ice stream" is removed and replaced with "upper Petermann Glacier" in the manuscript.*

P.19: Figure 3: change the unit for Fig. 3a/b colorbar from meters to millimeters. Also the caption does not need to say "from SICOPOLIS simulations for the year 1990". Also explain the label "NEEM zone" in the text. If you meant the NEEN ice core site, show the core site using a small symbol and explain it in the caption.

- *Changed the labels to mm here and on other figures.*
- *Removed "from SICOPOLIS…"*
- *Added a definition of NEEM zone as "NEEM zone is defined as area of the valley to the southeast of the NEEM ice core drilling site as indicated on Figure 1."*
- *Added a label for the NEEM ice core onto Figure 1b.*
- *Added "key locations and ice core sites" to the key of Figure 1.*

P.19 Figure 3c and 3d: consider adjusting the colorbar. The current figure does not support a statement in the main text about larger amount of water in the NEEM zone. The entire valley looks similarly red.

- *These figures show the basal water thickness difference so anywhere where it is yellow or red is where the water is thicker between the simulations.*

P.20 Figure 4: The labels of NEEM Zone for Figs. 3 and 4 point to different locations at a glance; adjust them so that both labels point to the right regions. Maybe adding arrows to Fig. 3 helps. Add the descriptions to the lower two panels (please clarify that these are zoom up of panels a and b). I understand that the authors chose the logarithmic scale because of the largely variable range of water flux shown here. But the river locations are shown in Fig. 3 already. Then is it actually better to show the water flux in the linear scale or at least adjust the colorbar so that the water flux increasing along the river can be better seen? I any case, clarify in the caption that the water flux colorbar is in the logarithmic scale if you stick with the logarithmic scale.

- *NEEM zone circles have been added to both figures to clarify the location.*
- *Added better NEEM zone labels to now Figure 7 (formerly Fig 4)*
- *Add c) and d) to the caption and stated that they are zoomed regions.*
- *Added "colours in a non-linear scale chosen to best demonstrate the distribution" to Fig 7.*
- *The water flux is not increasing along the river but is rather fairly constant.*

P.21 Figure 5: see the comment at P6L5ff.

- *Addressed earlier.*

P.22 Figure 6: Is it possible to show the extent of Petermann Ice Stream? The text says that the basal sliding increases significantly in the valley (> 10 m/a) but only little in the ice stream (< 0.5 m/a). It is hard to see the latter point in this figure. Maybe you can add the grounding line, or use non-white background color so that we can see near-zero difference (white) more clearly.

- *Added "Upper Petermann Glacier" label.*

P.23 Figure 7: see my comments above. And revise the unit for the basal water thickness colorbar from meters to millimeters.

- *changed scale to mm,*

P.24 Figure 8: Re-consider about this figure as suggested above (see my comments to p.8). If you retain Fig. 8 nearly as is, add colorbar, and the legend for blue/orange polygons. It is not easy to distinguish this orange polygon as the lower bed is shown with similar color. In any case, I expect that you modify Fig. 8 largely or delete it from the paper.

- *A colour bar has been added and the topography has been removed from the plot to make it clearer.*

Non-public comments to the Author:
Dear Chris, Ralf and others, I apologize that the review of this manuscirpt takes so long time. Also, I am afraid that you would be disappointed by seeing this large amount of revisions I am requesting in this stage. However, frankly speaking, the current manuscript is very hard to read and I think your messages can be much better presented if these points are considered. I hope to see your revision soon. Warmest regards, Kenny

- *Thanks again for the time you have spent reviewing this, we hope we have provided enough changes to progress forward and are happy to make further modifications accordingly.*
- *Best regards, the Authors.*

[revised manuscript text omitted]